# Evaluation of the current understanding of the impact of climate change on coral physiology after three decades of experimental research

Wiebke E. Krämer [1], Roberto Iglesias-Prieto [1,2] & Susana Enríquez [1✉]

After three decades of coral research on the impacts of climate change, there is a wide consensus on the adverse effects of heat-stress, but the impacts of ocean acidification (OA) are not well established. Using a review of published studies and an experimental analysis, we confirm the large species-specific component of the OA response, which predicts moderate impacts on coral physiology and pigmentation by 2100 (scenario-B1 or SSP2-4.5), in contrast with the severe disturbances induced by only +2 °C of thermal anomaly. Accordingly, global warming represents a greater threat for coral calcification than OA. The incomplete understanding of the moderate OA response relies on insufficient attention to key regulatory processes of these symbioses, particularly the metabolic dependence of coral calcification on algal photosynthesis and host respiration. Our capacity to predict the future of coral reefs depends on a correct identification of the main targets and/or processes impacted by climate change stressors.

[1] Laboratory of Photobiology, Unidad Académica de Sistemas Arrecifales (Puerto Morelos), Instituto de Ciencias del Mar y Limnología, Universidad Nacional Autónoma de México, Quintana Roo Cancún, Mexico. [2] Present address: Department of Biology, The Pennsylvania State University, University Park, PA 16802, USA. ✉email: enriquez@cmarl.unam.mx

Increases in atmospheric carbon dioxide concentrations derived from human activities are causing ocean temperatures to rise and ocean seawater-pH to decrease. Collectively, global warming and ocean acidification (OA) are considered the major global threats to marine ecosystems[1]. Coral reefs are particularly affected[2], as thermal anomalies of +1−2 °C above the regional maximum average in summer are considered the principal driver of severe loss in coral pigmentation[3,4] and symbioses function[5]. This phenomenon, known as coral bleaching, is responsible for massive coral mortality[4,6], with dramatic consequences for coral reefs[1,4,7]. Coral bleaching is predicted to increase in both severity and frequency as a result of climate change[6]. The impact of OA, resulting from the doubling of $pCO_2$ pre-industrial concentrations to 560 ppm, also predicts a 40% reduction in coral calcification by 2100[2,8]. However, while there is a large consensus on the impact of thermal stress for the induction of coral bleaching[4,5,9–11], the OA effects on coral physiology have not been clearly established (Table 1, Supplementary Information-SI, Table S1).

Coral bleaching is caused by a large accumulation of light-induced photodamage in the symbionts during heat-stress, evidenced by the reduction in the maximum photochemical efficiency ($F_v/F_m$), and it is preceded by severe loss of coral photosynthetic performance including reductions in coral pigmentation and symbionts[9–12]. Although reductions in $F_v/F_m$ and in coral pigmentation are commonly used as proxies for coral bleaching, the "bleached phenotype" only occurs at the end of the physiological disturbance[13,14], and expresses a dysfunctional condition of the symbiotic association, detectable, as previously proposed, by full suppression of coral photosynthesis[5,13]. The onset of heat-stress is determined by a temperature threshold, known as the "*Arrhenius break temperature*" (ABT)[15], above

**Table 1 Summary of literature data for the coral response to low seawater pH.**

**Effect of reduced-pH on coral performance**

| Article | $F_v/F_m$ | Photosynthesis | Calcification | Respiration | Symbiont density | Coral pigmentation |
|---|---|---|---|---|---|---|
| Agostini et al. 2013 | | No effect | | No effect | | |
| Anthony et al. 2008 | | − | − | No effect | | − |
| Bedwell-Ivers et al. 2016 | | − | − | − | No effect | No effect |
| Bedwell-Ivers et al. 2016 | | No effect | No effect | No effect | No effect | No effect |
| Camp et al. 2016 | | − | − | No effect | − | − |
| Castillo et al. 2014 | | | + | | | |
| Comeau et al. 2017 | | No effect | | No effect | | |
| Comeau et al. 2018 | | | − | | | |
| Cornwall et al. 2018 | | | No effect | | | |
| Crawley et al. 2010 | | − | | + | No effect | |
| Davies et al. 2016 | | | | + | | |
| Enochs et al. 2014 | No effect | | No effect | | | |
| Godinot et al. 2011 | No effect | No effect | | No effect | No effect | No effect |
| Hii et al. 2009 | | − | − | − | − | |
| Hii et al. 2009 | | + | + | + | − | |
| Hoadley et al. 2016 | + | No effect | | | No effect | |
| Hoadley et al. 2016 | No effect | No effect | | | No effect | |
| Horwitz and Fine 2014 | | | | | − | − |
| Horwitz and Fine 2014 | | | | | No effect | No effect |
| Houlbreque et al. 2012 | No effect | No effect | No effect | No effect | No effect | No effect |
| Kanieswska et al. 2012 | | − | No effect | − | − | |
| Krief et al. 2010 | | | − | | − | |
| Krueger et al. 2017 | No effect | + | No effect | No effect | No effect | |
| Langdon and Aktinson 2005 | | + | − | | | |
| Leclercq et al. 2002 | | No effect | − | + | | |
| Marubini et al. 2008 | − | No effect | | | No effect | |
| Mason 2018 | | | | | − | |
| Noonan and Fabricius 2015 | + | + | | No effect | | + |
| Noonan and Fabricius 2015 | + | No effect | | No effect | | No effect |
| Ogawa et al. 2013 | No effect | | | | No effect | |
| Ohki et al. 2013 | No effect | | − | | | |
| Reynaud et al. 2003 | | − | | No effect | + (CSD) | No effect |
| Rodolfo-Metalpa et al. 2011 | | | + | | | |
| Rodolfo-Metalpa et al. 2010 | | No effect | No effect | No effect | + | + |
| Rodolfo-Metalpa et al. 2010 | | No effect | No effect | No effect | No effect | No effect |
| Schoepf et al. 2013 | | | − | | No effect | − |
| Schoepf et al. 2013 | | | No effect | | − | No effect |
| Schoepf et al. 2013 | | | No effect | | No effect | No effect |
| Suggett et al. 2013 | | + | − | No effect | | |
| Takahashi et al. 2013 | | No effect | No effect | No effect | No effect | |
| Tremblay et al. 2013 | | − | | No effect | − | − |
| Wall et al. 2017 | | | No effect | | No effect | No effect |
| Suggett et al. 2012 | No effect (anemone) | + (anemone) | | + (anemone) | + (anemone) | |
| Towanda and Thuesen 2012 | | + (anemone) | | + (anemone) | + (anemone) | |

Results published in 36 studies documenting no effect, positive (+) or negative (−) significant effects for different coral traits. See Supplementary Information for references.
CSD cell specific density.

which coral photosynthesis[5,16,17] and calcification[5,16,18] diminish with temperature. Such a tipping point has not been documented for coral respiration at this temperature[5]. Below ABT, exposure to elevated temperatures is generally beneficial for all metabolic rates, as they accelerate the enzymatic processes. Values for ABT and for the $Q_{10}$ temperature coefficient (i.e., the factor by which the rate of a metabolic process increases for every 10-degree rise in temperature) of symbiotic corals are variable among species, metabolic processes, and the acclimatory phenotype of the organisms[5].

The coral response to OA is less understood[19] (Table 1, SI-Table-S1). Absorption of $CO_2$ by the ocean surface modifies seawater chemistry leading to a reduction in pH and aragonite saturation state $(\Omega_{arag})$[19]. Initially, OA was documented to affect the biomineralization process in a wide range of marine calcifying organisms[20], including planktonic coccolithophorids[20], foraminifera[20], crustaceans[21], molluscs[21], coralline algae[21–23], and corals[23–25]. Further studies, however, started to question such adverse effects of OA on marine calcification[26,27]. Changes in $\Omega_{arag}$ were found to be positively correlated with the decline in coral calcification[28], but later this decline was found to be associated with changes in seawater-pH and $pCO_2$ rather than $\Omega_{arag}$ per se[29]. Other studies have reported no reductions in coral calcification or even its stimulation under OA conditions (Table 1, SI-Table-S1). A meta-analysis concluded almost a decade ago that OA does not affect coral photosynthesis, and identified a wide species-specific component of this response[30]. Experimental analyses of the combined effect of elevated temperature and OA have reported a large diversity of coral responses[31–33], with an inconsistent role of temperature in the modulation of the OA impact[17,25,33,34]. More recently, a new meta-analysis has concluded that the additional effect of OA under intensifying marine heatwaves will lead to a greater impact on photosynthesis and coral survival[35]. Unfortunately, a limited number of studies have characterized photosynthesis and calcification rates *in simultaneum*, despite the known dependence of coral calcification on photosynthetic products such as glycerol, glucose and oxygen[36,37]. Our literature review unveils the incomplete analyses and partial views of most of the experimental characterizations (Table 1), which could explain the still insufficient knowledge of the OA effects on coral physiology.

It has been hypothesized that OA may increase coral photosynthesis rates by inducing moderate enhancements in $pCO_2$, postulating that carbon availability limits both coral calcification and photosynthesis[38], a controversial assumption, however, not yet demonstrated. While slight increases in net photosynthesis have been reported, most studies have concluded that coral photosynthesis is in general unaffected by OA (Table 1). A potential enhancement in algal photosynthate translocation under lower pH has been also suggested[39]. Contrary to a positive or neutral impact, it has also been concluded that elevated $pCO_2$ can induce coral bleaching in corals as well as crustose coralline algae[23]. However, this conclusion may be related to high experimental levels of light-stress during the experiment, as demonstrated for coralline algae[22]. In this latter study, the authors also documented larger impacts of heat-stress to algal photosynthesis and calcification than those of OA, and that the individual effect of OA on coralline physiology is independent of the photosynthetic process[22], in agreement with previous findings for corals[33]. In addition, the OA adverse effects have been also explained through the impact of light and photosynthesis on algal energetic budgets and resource allocation[40] or through the capacity of light to modulate the susceptibility of coral calcification to OA, considering the high sensitivity of corals to light-stress under elevated temperatures[41].

To resolve these diverse and often contradictory interpretations, we investigated here the physiological response of four Caribbean reef-building species, *Pseudodiploria strigosa*, *Orbicella faveolata*, *Montastraea cavernosa,* and *O. annularis*, to the direct and combined effects of elevated temperature [+2 °C above the photoacclimatory condition of the experimental organisms, which is the local maximum of 30 °C in summer[5]; and low-pH [pH = 7.9; equivalent to a $pCO_2$ of 887 μatm], which represents the expected changes by 2100 (scenario-B1 in IPCC 2007; and SSP2-4.5 in 2021)[42,43]. As a prerequisite to elucidate the direct physiological impact of each stressor on coral performance, the experiment was fully factored designed for temperature and $pCO_2$, and maintained moderate levels of light stress throughout the experiment. We characterized the variation in coral physiology, coral optics, and structural traits at days 0 and 10. Daily changes in solar radiation, the magnitude of photodamage accumulation, and light absorption during the experiment were also monitored. Photodamage accumulation was quantified by daily dusk measurements of the maximum photosystem II, PSII, and photochemical efficiency ($F_v/F_m$).

## Results

### Impact of experimental treatments on photodamage accumulation in the symbionts.
Exposure to heat-stress produced large and progressive $F_v/F_m$ declines in both treatments, whereas no significant change was observed in the OA treatment (Fig. 1; SI-Table-S2). Photodamage accumulation in the symbionts varied from 22% reduction in $F_v/F_m$ for *O. faveolata*, to 61% for *M. cavernosa*. A positive and significant effect of OA on $F_v/F_m$ was found for *P. strigosa*, which also showed a significant interaction with heat-stress, as $F_v/F_m$ was higher in the combined treatment (Fig. 1b; SI-Table S2). Significant $F_v/F_m$ recovery was observed in *O. annularis* during overcast days (days 6–10; Fig. 1e), whereas no beneficial effect was detected in *M. cavernosa* (Fig. 1d).

### Impact of treatments on coral metabolic rates and optical and structural traits.
Coral metabolic rates (maximum gross photosynthesis, $P_{max}$; light-enhanced respiration, $R_L$; and maximum calcification, $G_{max}$) and structural traits (symbiont and chlorophyll *a*, Chl*a*, density and symbiont cell pigmentation, $C_i$) showed no changes after exposure to control conditions. Only *O. annularis* presented a slight increase in $C_i$ together with small reductions in calcification (SI-Table S3). For the OA treatment, changes were mostly insignificant. Only *P. strigosa* presented a significant increase in $P_{max}$ at control temperature (interactive effect of low pH and temperature, two-way ANOVA, $p < 0.05$, SI-Table S4, Fig. 2) and reduced calcification rates at low pH (two-way ANOVA, $p < 0.05$, SI-Table S4, Fig. 2), whereas Chl*a* density and the ratio $P_{max}$ to $R_L$, P/R, were reduced in *O. annularis* (Fig. 2, two-way ANOVA, $p < 0.05$, SI-Table S4). Symbiont density at low pH and control temperature was slightly, but not significantly increased in *P. strigosa* and reduced in *O. annularis* (Fig. 2). Exposure to heat-stress, however, resulted in large declines in coral metabolic rates, pigmentation, and absorptance (Fig. 2; SI-Tables S4, S5). Loss of pigmentation was particularly important for *O. annularis*, which showed the largest reduction in Chl*a* density (78%), whereas the smallest reductions (57%) were measured for *P. strigosa* (Fig. 2). Decreases in coral pigmentation were mainly due to symbiont loss for *P. strigosa* (36%), *O. faveolata* (52%) and *O. annularis* (69%). In *M. cavernosa*, such decrease was due to a 50% reduction in symbiont $C_i$, as symbiont losses were insignificant (Fig. 2K). Consistent with the large declines in $F_v/F_m$ and coral pigmentation, all heat-stressed organisms showed significant reductions in $P_{max}$ (Fig. 2;

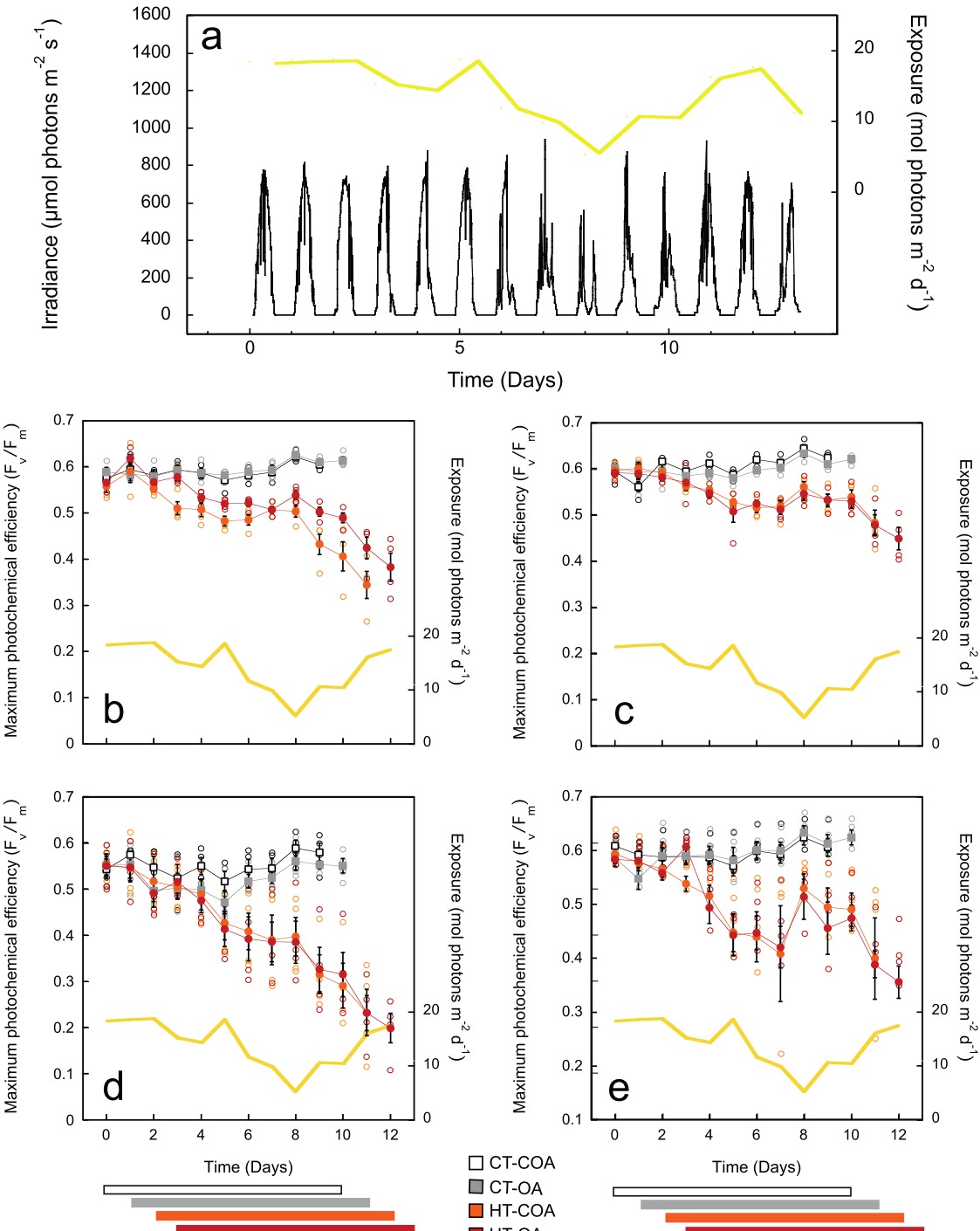

**Fig. 1 Variation in light conditions and maximum photochemical yield of PSII ($F_v/F_m$) throughout the experiment. a** Variation in irradiance (μmol quanta m$^{-2}$ s$^{-1}$), in black, and in diurnal light exposure (mol quanta m$^{-2}$ d$^{-1}$), in yellow, at the level of the experimental coral nubbins, over the experimental period. Variation in $F_v/F_m$ for: **b** *Pseudodiploria strigosa*; **c** *Orbicella faveolata*; **d** *Montastraea cavernosa*; and **e** *Orbicella annularis*, exposed to four treatments: CT-COA—ambient temperature-ambient pH; CT-OA—ambient temperature-low pH; HT-COA—high temperature-ambient pH; HT-OA—high temperature-low pH. Values displayed are the mean ± SE of four replicates ($n = 4$). The horizontal bars at the bottom of the figure illustrate the beginning and end of each 10-day treatment. Empty circles represent the values for each sample.

SI-Tables S4, S5). However, full suppression of photosynthesis (91–95% reductions, with final values not different from zero; t-test, $p > 0.05$) was only measured for *P. strigosa* and *M. cavernosa*, whereas *O. annularis* still maintained 13% and 6% of the control photosynthetic activity in the two heat-stress treatments, despite its large Chl*a* and symbiont losses (Figs. 2 and 3). The most tolerant species to heat-stress was *O. faveolata*, which was able to maintain

33% of $P_{max}$ after 10 days of stress exposure (Figs. 2 and 3). Significant adverse impacts of heat-stress were also observed on the respiration rates of *M. cavernosa* (42%) and *O. annularis* (36%; Fig. 2; SI-Table S5). The ratio of $P_{max}$ to $R_L$ (P/R) ranged from 1.96 in *O. annularis* to 2.3 in *O. faveolata*, and did not change during the experiment in the control treatment and with increasing levels of $CO_2$ under control temperature (Fig. 2). However, P/R showed

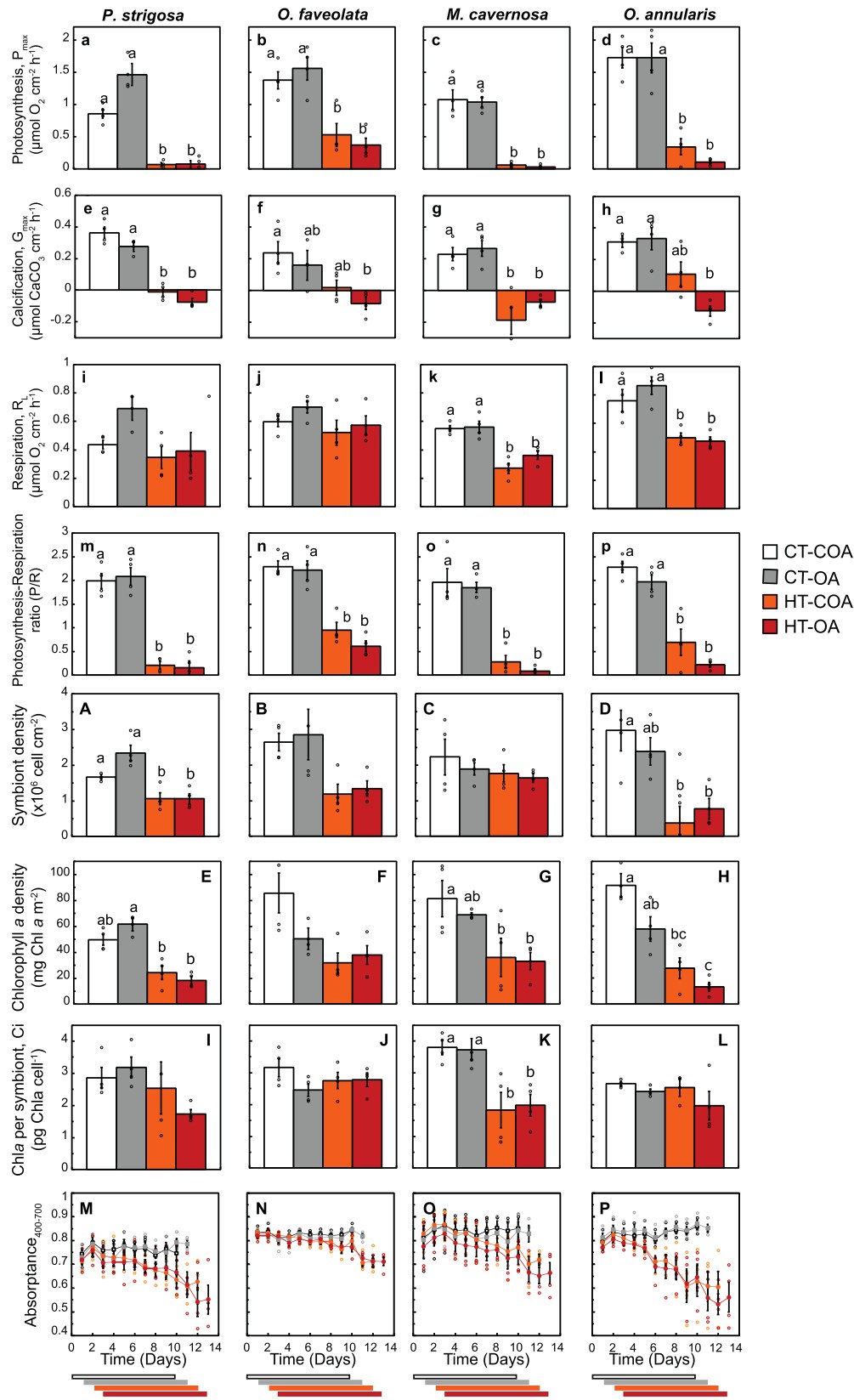

dramatic reductions in all heat-stressed corals, with values significantly lower than 1 (Fig. 2; SI-Tables S4, S5). The largest reductions (91–80%) were measured for *P. strigosa*, *M. cavernosa*, and *O. annularis*. *O. faveolata*, with a 66% reduction, could maintain P/R values not significantly different from 1 after 10 days of exposure to heat-stress.

**Understanding the impact on coral calcification and the diversity of coral responses to OA**. Coral calcification was strongly impaired in all species after exposure to heat-stress (Figs. 2 and 3), with full suppression of calcification at the end of both experimental treatments (values not different from zero; t-test, $p > 0.01$). For *M. cavernosa*, there was significant decalcification

**Fig. 2 Physiological, structural and optical coral responses to the experimental treatments.** Physiological (**a–p**), structural (**A–L**), and optical (**M–P**) responses of *Pseudodiploria strigosa*, *Orbicella faveolata*, *Montastraea cavernosa*, and *Orbicella annularis* to the experimental treatments. Variation (mean ± SE; $n = 4$) in **a–d** maximum gross photosynthesis, $P_{max}$; **e–h** maximum calcification, $G_{max}$; **i–l** light-enhanced respiration, $R_L$; **m–p** photosynthesis-respiration ratio, P/R; **A–D** symbiont density; **E–H** chlorophyll *a* density, Chla; **I–L** algal Chla cell content, Ci; and **M–P** coral Absorptance (400–700 nm); for corals exposed for 10 days to control (CT-COA—Ambient temperature-ambient pH; in white), low pH (CT-OA—Ambient temperature-low pH; in gray); heat-stress (HT-COA—high temperature-ambient pH; in orange); and combinations of elevated temperature and low pH (HT-OA—High temperature-low pH; in red). Different letters indicate significant differences between treatments determined using Tukey HSD post hoc tests ($p < 0.05$). The horizontal bars at the bottom of the figure illustrate the beginning and end of each 10-day treatment. Empty circles represent the values for each sample.

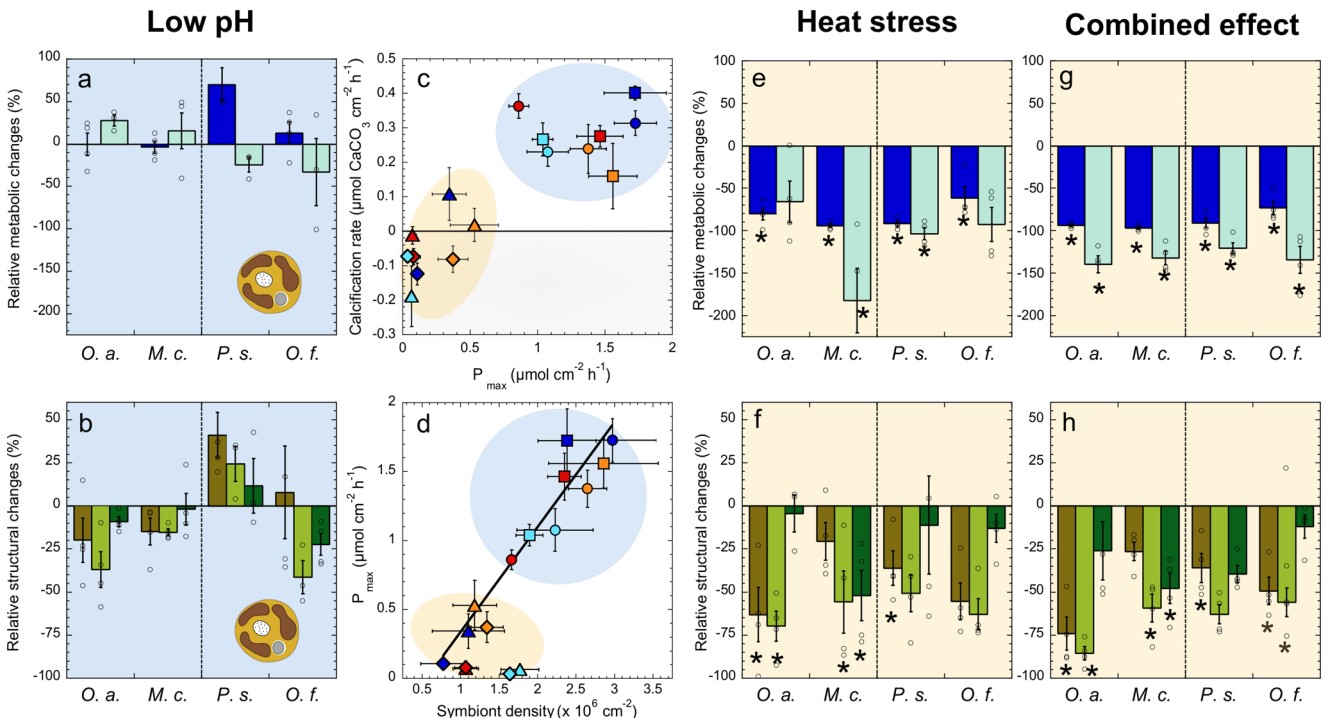

**Fig. 3 Comparison of the impact of ocean acidification (OA) heat-stress and combined effect on *Orbicella annularis*, *Montastraea cavernosa*, *Pseudodiploria strigosa*, and *Orbicella faveolata*.** Values (means ± SE; $n = 4$) are expressed as relative changes with respect to the control. The left panels in blue shade (**a**, **b**) illustrate the effect of OA (CT, OA) on coral physiology (gross photosynthesis, $P_{max}$ [blue], and calcification, $G_{max}$ [mint green]), and coral pigmentation (symbiont [khaki], Chla [green] density, and symbiont cell pigmentation, Ci, [dark green]). The right panels in yellow shade (**e–h**), describe relative changes in the physiological (**e**, **g**) and structural coral traits (**f**, **h**) in response to: **e**, **f** heat-stress (HT-COA) and **g**, **h** the combined effect of heat-stress and ocean acidification (HT-OA). Values marked with Asterisks (*) indicate significant differences against the control (CT-COA) determined using Tukey HSD post hoc tests ($p < 0.05$). Middle panels (**c**, **d**) illustrate the differential impact of OA (blue shade) and heat-stress (yellow shade) on the association of variation between: **c** coral calcification, $G_{max}$, (µmol $CaCO_3$ cm$^{-2}$ h$^{-1}$) and gross photosynthesis, $P_{max}$ (µmol $O_2$ cm$^{-2}$ h$^{-1}$; and **d** between $P_{max}$ and changes in symbiont density (×10⁶ cells cm$^{-2}$). Circles represent the CT-COA treatment; squares—CT-OA; triangles—HT-COA; diamonds—HT-OA, for *O. annularis* (dark blue, *O.a.*), *M. cavernosa* (blue turquoise, *M.c.*), *P. strigosa* (red, *P.s.*) and *O. faveolata* (orange, *O.f.*). Empty circles represent the values for each sample.

activity after exposure to the combined treatment (Fig. 2; SI-Table S5). A statistically significant additive interaction between heat-stress and OA for coral calcification was only found for *O. annularis* (140% reduction; two-way ANOVA, $p < 0.05$; Figs. 2 and 3; Table S4), resulting in substantial carbonate dissolution activity in both heat stress treatments. Globally, the impact of OA alone on coral calcification was less severe than that found for heat-stress and more variable among species with no consistent pattern (Fig. 3). Using a PCA, we identified two types of responses to OA (Fig. 4a; SI-Table S6). The first was represented by the eight samples analyzed of *O. annularis* and *M. cavernosa* (i.e., 4 replicates per species), and one sample of *O. faveolata*. The second was represented by all four samples of *P. strigosa* and two samples of *O. faveolata*. (Fig. 4a). The fourth replicate of *O. faveolata* showed very low values in all descriptors, suggesting a particular low performance for this sample independently of the treatment applied. According to this variability, the first group was characterized by

slight increases in $G_{max}$ (21.8% ± 10.2; t-test = 2.5; df = 8; $p < 0.006$) and no change in $P_{max}$ despite small reductions in pigmentation and symbiont content (−26.1% ± 6.4 and −17.3% ± 7.0, respectively; Fig. 4b; t-tests = −4.7; −2.99; df = 8; $p < 0.02$). Such decreases in pigmentation under low pH, however, were not significant for any species, when comparing the variability among species (Figs. 2, 3B; SI-Table S5). For the second cluster defined by the PCA formed by *P. strigosa* and two samples of *O. faveolata*, the effect of OA resulted in increases in symbiont content and $P_{max}$ (Fig. 4b; 43.7% ± 11.8 and 57.2% ± 12.1, respectively, t-tests = 3.7, 4.7; df = 5; $p < 0.02$), whereas $G_{max}$ showed slight reductions (−26.8 ± 4.2%; t-test = −6.4; df = 5; $P < 0.002$; Fig. 4b). The large variability showed by *O. faveolata* indicates that the four replicates used in this analysis were insufficient to characterize its OA response. Non-significant changes in symbiont density were estimated for this species, although both Chla density and $C_i$ decreased slightly (Fig. 2).

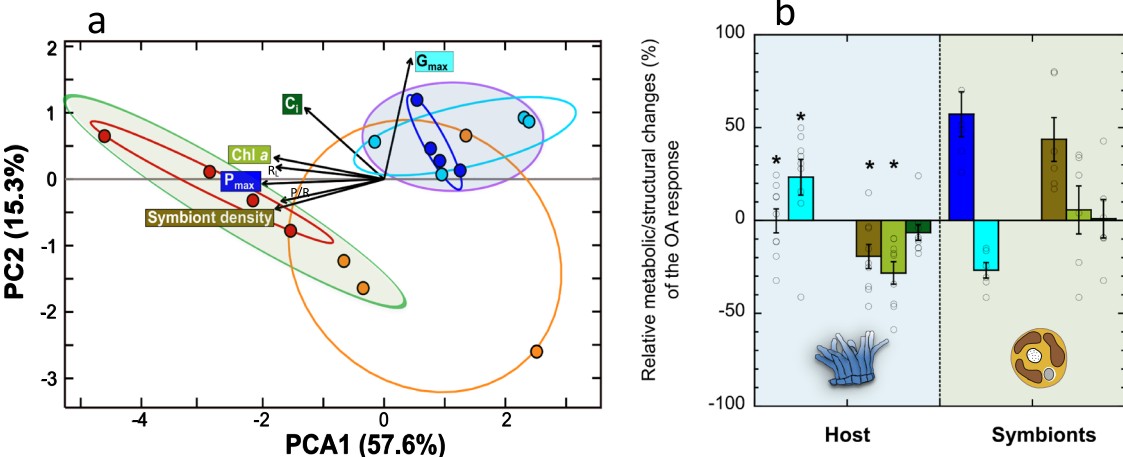

**Fig. 4 Two types of coral responses to ocean acidification (OA).** In panel **a**, the PCA analysis illustrates the two responses identified clustering all samples ($n = 4$ per species) into two groups with contrasting responses. The first group (blue shade) was represented by *O. annularis* (dark blue circles), *M. cavernosa* (blue turquoise circles) and one sample of *O. faveolata* (orange circles). The second group (green shade) was represented by *P. strigosa* (red circles) and two samples of *O. faveolata* (orange circles). A fourth sample of *O. faveolata* showed very low values in all measured descriptors, indicating low performance for this particular sample. Panel **b** shows averages ± SE for relative changes in gross photosynthesis, $P_{max}$ [blue], calcification, $G_{max}$ [mint green]), symbiont [khaki], Chla [green] density, and symbiont cell pigmentation, Ci, [dark green]), for all coral samples of each of the groups identified in the PCA analysis. The first group in blue shade, showed a positive response for host calcification together with slight reductions in symbiont content and holobiont pigmentation. The second group, in green shade, showed positive responses for algal photosynthesis and symbiont content. Asterisks (*) indicate significant differences (t-test; $p < 0.05$) between the two groups for $P_{max}$ (t-test = −4.6, df = 13, $p < 0.001$), $G_{max}$ (t-test = 4.23, df = 13, $p < 0.001$); symbiont (t-test = 5.1, df = 13, $p < 0.001$) and pigment density (t-test = −2.7, df = 13, $p < 0.02$) density. Empty circles represent the values for each sample.

## Discussion

The results support that heat-stress induces more adverse effects on coral physiology and pigmentation than OA, in agreement with a previous study[33]. Applying a thermal anomaly of +2 °C for 10 days, we observed severe disturbances on coral photosynthesis and calcification, whereas the experimental simulation of the expected OA conditions by 2100 [scenario-B1[42] or SSP2-4.5[43]] caused moderate changes in coral performance. Accordingly, global warming represents a more significant threat for coral calcification than ocean acidification, particularly considering that the adverse impacts of heat-stress occur substantially earlier than the 'bleached coral phenotype' was developed, in accordance with the definition of coral bleaching recently proposed[13,14]. The OA interaction with heat-stress was, comparatively, a minor factor in our analysis, in disagreement with previous conclusions[35]. Positive OA effects on $F_v/F_m$ acting to dampen the reduction caused by heat-stress have been formerly documented in corals[44] and corallimorphs[45]. We observed this beneficial effect only for *P. strigosa*, but such dampening of $F_v/F_m$ reduction was not mirrored in the photosynthetic rates.

The large repertoire of coral traits analyzed allowed an ample description of the physiological response of the four species investigated. In addition, the use in our analysis of moderate levels of light and corals already acclimatized in situ to the local summer maximum of 30 °C were fundamental conditions to minimizing interference from other stress-factors or physiological adjustments. The low $F_v/F_m$ variation measured for the control corals confirms the appropriate acclimatization of all species to the experimental tank conditions. In contrast, the rapid accumulation of photodamage during heat-stress, a confident indicator for light-stress, highlights its central role in the heat-stress response of these symbioses. These results agree with previous findings for corals[5,9–12] and coralline algae[22]. However, the magnitude of $F_v/F_m$ declines did not mirror the actual impact on coral photosynthesis or the magnitude of pigment or symbiont losses. For example, a complete cessation of photosynthesis was

found at an intermediate state of photodamage accumulation, as assessed by $F_v/F_m$ for *P. strigosa*, and at the lowest symbiont losses for *M. cavernosa*. Conversely, *O. annularis* still maintained some photosynthetic activity despite the large photodamage accumulation, and the fact that this species exhibited the largest reductions in coral pigmentation and symbionts. These findings question the suitability of coral pigmentation as a proxy for coral bleaching, and $F_v/F_m$ as a single measure of impact on photosynthesis, two parameters commonly used to identify the bleached coral phenotype and to quantify stress impacts on corals. Likewise, it underlines the need of a better proxy for the description of the dysfunctional condition of the symbiotic association.

Similar substantial impacts of heat-stress on coral photosynthesis and calcification were observed in the four species, despite the potential genetic variability of the dominant symbiont types[46,47]. Two species, *O. annularis* and *M. cavernosa*, were more sensitive to light-stress, whereas *O. faveolata* was more tolerant to heat-stress. Hence, our study confirms the close dependence of the impact of heat-stress on the prevailing levels of solar irradiance[5,48–50], and stresses the relevance of symbiont capacity *in hospite* to cope with light-stress to explain coral susceptibility to heat-stress[51,52]. Interestingly, beneficial effects of overcast days on coral recovery were only measured for *O. annularis* (Fig. 1e). In this species, the symbionts of the "heat-stressed" samples were still able to release some photosynthates and oxygen to the host, which may explain why the ability to recover was still maintained in this species, and absent in the species that had already developed the 'bleached' phenotype[14]. Our results do not support the conclusion that the level of light stress "sensed" by the algae *in hospite*, regulates the susceptibility of coral calcification to acidified conditions[41] or that OA can induce coral bleaching[23]. However, they support the importance of discerning between the dysfunctional ("bleached") and the "stressed" coral phenotypes, as previously proposed[5,12,14].

Positive and negative effects of OA on coral calcification, photosynthesis and symbiont and pigment content have been

documented, according to our literature review, as have the absence of any impact[29,53–56] or even positive synergistic effects of the interaction with heat-stress[23,33] (Table 1, SI-Table S1). This review includes analyses on tropical and temperate species (58 and 8 studies, respectively). $F_v/F_m$ and coral photosynthesis are generally unaffected[30], despite some reports of minor increases due to OA. Some of these findings may not reflect the direct effect of OA on photosynthesis, as most studies report net photosynthetic rates, which cannot differentiate between respiration and photosynthesis impacts. The majority of this research is focused on coral calcification (54 studies), and only 30 studies (including anemones) document changes in photosynthesis, while only 13 characterize changes in symbiont and pigment content. In our review, only 19 studies measured OA effects on both coral calcification and photosynthesis, and even fewer, 5 studies, have considered the evaluation of the OA effect on symbiont and pigment content. Hence, this literature evaluation unveils the still incomplete physiological characterization of the coral response to OA. Likewise, our experimental analysis indicates that insufficient attention to coral respiration and to the metabolic-dependence of coral calcification on photosynthesis and respiration has also affected the understanding of this response. Coral biomineralization in addition to inorganic carbon requires oxygen and the energy supplied by both, algal photosynthesis and host respiration[36,37]. We observed increases in coral respiration, although not significant, at an enhanced DIC supply (dissolved inorganic carbon, OA treatment), an observation consistent with previous findings and with a transcriptomic analysis[57]. Furthermore, heat-stress caused severe reductions in the photosynthesis to respiration ratio (P/R) of all species, which may have led to a differential energy shortage (ATP synthesis) and reductions in oxygen delivered to support coral calcification[36,37]. Hence, attention to the variation in coral respiration and to the metabolic-dependence of coral calcification on photosynthesis and respiration rates, is key for disentangling this response under stress. In fact, coral respiration is not only an important source of metabolic energy (ATP) and respiratory carbon for maintaining coral calcification, but also a source of reactive oxygen species, ROS[58]. Moreover, the control of cellular redox balances and ATP synthesis relies on mitochondria[59].

It has been postulated that an increase in DIC supply under acidic conditions may favor coral calcification and photosynthesis. Evidence has been reported both in support of DIC limitation of coral photosynthesis[57,60,61] and no limitation[62,63]. The finding of a large species-specific variability has led to the interpretation that DIC may have an environmentally unregulated control or the occurrence of an 'intentional' DIC limitation by the host. Our study cannot provide any insight into the biological mechanisms, but certainly uncovers a substantial species-specific component, and/or related to the coral condition/phenotype, of the coral response to OA. We could identify in this study two types of responses. One, represented by *O. annularis* and *M. cavernosa* and that also included a sample of *O. faveolata*, was characterized by slight enhancements in calcification and no impact on photosynthesis despite small reductions in symbiont and pigment content. The other, represented by *P. strigosa* and two samples of *O. faveolata*, was characterized by increases in symbiont density and photosynthesis, with minor adverse effects on calcification after 10 experimental days. Biological processes such as the downregulation of energetically expensive carbon concentration mechanisms and/or the reallocation of this energy into different 'metabolic routes', such as calcification or photosynthesis could explain this variability. In support of this interpretation, a recent transcriptomic analysis[64] has documented the downregulation of genes involved in carbon acquisition by the symbionts under elevated $p$CO$_2$. The present comparative

analysis cannot explain the 'variable' coral response to OA observed, but allowed highlighting important gaps in the knowledge of the regulation of these symbioses that still need to be elucidated in order to understand it. This variability may have a species-specific component, as well as be regulated by the physiological condition of the holobiont (i.e., phenotype), as observed for *O. faveolata*.

It has been argued that only meta-analyses can contribute to global projections of climate change impacts on coral reefs at the end of the 21st century[65], as experimental analyses cannot reproduce the complexity of interactions that occur in nature. Global projections using meta-analysis, however, have concluded that OA will cause significant additional impacts under intensifying marine heatwaves[35]; or that the large declines predicted in net carbonate production of coral reefs are not direct impacts on organism's calcification rates or bioerosion, but reduction in coral cover[66]. Our study does not support these conclusions, and show a minor effect of OA on coral calcification. Studies on carbonate dissolution, have also documented that OA will affect coral reefs through an increase in net sediment dissolution[67–69]. As the capacity to predict the future of coral reefs depends on a correct identification of the main targets and/or processes impacted by climate change stressors, experimental analyses cannot be neglected, as they are essential in isolating and thus elucidating both the individual impacts and possible interactions of environmental factors on organism performance. Nonetheless, the success of experimental studies depends on the level of knowledge of the system under analysis. Hence, a better understanding of the entirety of the physiological and cellular processes evoked or compromised under stress is fundamental to improve our capacity to predict the impact of climate change on the main reef builders.

In summary, our study supports that global warming represents a more significant threat to coral growth and reef accretion than ocean acidification. Heat-stress directly affects coral performance through *in hospite* exacerbation of light-stress in the symbionts, whereas ocean acidification induces moderate effects on coral metabolism, some of them even positive. The impact of heat-stress on coral calcification primarily relies on the dependence of the biomineralization process of algal photosynthetic products, a process traditionally known as 'light-enhanced-calcification'[60,70]. However, coral respiration also plays an important role as another source of carbon and metabolic energy for coral calcification that needs to be understood. We hypothesize here that two contrasting pathways in holobiont carbon allocation, and in the flexibility of the regulation of the metabolic link between algal photosynthesis and host calcification, may contribute to explaining the large variability of moderate effects induced by ocean acidification on coral physiology. A better knowledge of this regulation is fundamental for elucidating the impact of climate change on symbiotic corals, and it may also allow explaining why a robust species to heat-stress such as *Orbicella faveolata*, also presents one of the largest adverse impacts of global warming on calcification.

## Methods

**Sample collection**. Fragments of three colonies of *Pseudodiploria strigosa*, *Montastraea cavernosa*, *Orbicella annularis*, and *O. faveolata* were collected at a depth of 4–5 m in the back-reef of Puerto Morelos, Mexico (20°54'19.80"N, 86°50'6.20" W). Coral fragments were transferred to the UNAM mesocosm system and located in outdoor tanks with running seawater from the lagoon. Organisms were kept under natural solar irradiance dampened to 47% of the surface irradiance ($E_s$) using neutral screening, which corresponded to the light intensity at the sampling depth. One day after sampling, corals were cut into smaller pieces of roughly equal size (2 cm$^2$). Experimental organisms were allowed to recover for two days before gluing them onto PVC plates. For full recovery, coral nubbins were fixed onto tables, which were placed in the lagoon at 4 m depth. The nubbins used in this experiment were maintained in the reef lagoon of Puerto Morelos for 1 year.

**Experimental conditions**. We followed a similar protocol applied in Vasquez-Elizondo and Enríquez[22], at the UNAM mesocosm facilities of Puerto Morelos (see below). Natural variation in solar irradiance during the experiment presented an average in diurnal light exposure of $14.1 \pm 1.14$ mol quanta $m^{-2} d^{-1}$. This variability simulated the natural light conditions to which organisms were pre-acclimated and recovered "in situ" in the reef lagoon of Puerto Morelos after their manipulation. From day 6 until day 10 of the experiment, we measured a 60% reduction in light exposure due to very dense cloud cover (Fig. 1a). The temperature in the control tanks (CT) was $29.93 \pm 0.19\,°C$ and corresponded to the local maximum mean in summer in August[5], and the temperature of acclimatization of the experimental organisms "in situ". The high-temperature treatment (HT; average of $31.93 \pm 0.25\,°C$)[22], represented $+2\,°C$ above the local summer maximum.

Seawater chemistry in the experimental tanks showed aragonite saturation state values ($\Omega_{arag}$) ~40% lower ($\Omega_{arag}$ = 2.29–2.44) in the OA treatment (OA; pH = 7.9) than at ambient control-pH (COA; pH = 8.1; $\Omega_{arag}$ = 3.53–4.18; SI-Table-S7). The concentration of $HCO_3^-$ increased from ~1653 mol $kg^{-1}$ at pH = 8.1 to 1943 mol $kg^{-1}$ at pH = 7.9, and $CO_2$ concentrations increased from ~387 to 887 µatm. In contrast, the average $CO_3^{2-}$ concentration decreased from 217 to 253 µmol $kg^{-1}$ in the control/ambient treatment to 140–148 µmol $kg^{-1}$ in the OA, pH = 7.9, treatment (SI-Table-S7). Total alkalinity was not affected by the reduction in seawater pH, remaining within the range of 2277–2320 (µmol $kg^{-1}$).

**Experimental design**. To test the effects of heat-stress and OA, we used a fully orthogonal, two-factor design. Corals were maintained in a water table with a seawater flow-through system. Experimental 30-L tanks were supplied with constant seawater from four 1000-L header tanks, which had access to direct seawater flow from the reef lagoon. The average flow rate in the experimental tanks was $0.33\,L\,s^{-1}$ with a turnover rate of about 90 s. Corals were exposed to experimental conditions in 8 tanks ($n = 2$ treatment$^{-1}$, 2 coral nubbins species$^{-1}$ tank$^{-1}$) for 10 days in August–September 2013. The temperature treatments were set based on the local summer maximum of the reef lagoon of Puerto Morelos (30 °C)[5] achieved in August (ambient control temperature—CT), and a thermal anomaly of $+2\,°C$ above this summer average (32 °C) considered the heat-stress treatment (HT). Water temperature was independently regulated using titanium immersion heaters (Process Technology, Ohio, USA) in the main reservoirs, which supplied seawater to the experimental tanks. For the OA-treatments, the current seawater pH in the lagoon (pH 8.1) was used as a control condition (ambient pH – COA) and the changes expected by 2100 under the carbon emission scenario B1[42] or SSP2-4.5[43], with pH units in the National Bureau of Standards (NBS) scale of pH = 7.9, were used to determine the OA-treatment (OA). The seawater carbonate system of the OA treatments was automatically adjusted by bubbling with $CO_2$ using electronic valves (Sierra Instruments, INC, Smart-Track 2) until reaching the desired pH. Well-mixed water was continuously pumped from the main reservoir to the corresponding experimental tanks. Water temperature and pH were continuously monitored using a pH-electrode (resolution of 0.01 pH units; Thermo Scientific, Inc., USA) and a thermocouple-based custom-made probes (J-types; resolution of 0.1 °C; TEI-Ingeniería, Mexico), respectively, both connected to a computer-based system equipped with a wireless data acquisition card (National Instruments, Texas, USA). Throughout the experiment, we used natural solar irradiance and neutral screening to maintain the pre-acclimatory conditions (47% $E_s$). Variation in solar irradiance was monitored using a cosine-corrected sensor LI-192 quantum sensor (LI-COR, Lincoln, USA) connected to a data logger (LI-COR LI-1400, Lincoln, USA) and registered in 5-min intervals at the pier of the UASA. Due to the time required to perform the physiological determinations of four species, each treatment was initiated with a delay of one day in a progressive manner. This allowed finishing all physiological determinations of one treatment in one day, while maintaining the same duration of each treatment for all organisms. The experiment was started with the control treatment (CT-COA), followed by the CT-OA treatment on day 1, the HT-COA on day 2 and ultimately, HT-OA on day 3.

**Seawater carbonate chemistry**. Throughout the experiment, seawater carbon chemistry in the reservoirs was monitored every second day at solar noon (~1 p.m.). Values of $pH_{NBS}$, salinity, temperature, and total alkalinity (AT, µmol $kg^{-1}$) were used to calculate the components of the carbonate system in seawater ($pCO_2$, $CO_3^{2-}$, $HCO_3^-$, DIC concentrations, and $\Omega_{arag}$) using the CO2SYS software in Microsoft Excel[71]. Validation of AT values was performed using certified reference material from the Andrew Dickson's lab (Scripps Institution of Oceanography, USA).

**Chlorophyll *a* fluorescence**. Daily variation in the maximum photochemical efficiency of PSII ($F_v/F_m$) of Symbiodininaceae *in hospite* for each coral nubbin was recorded at dusk (7:30 p.m.–8 p.m.) using a DIVING PAM fluorometer (Walz GmbH, Effeltrich, Germany), following the same protocol used in Scheufen et al.[5]. Variation in $F_v/F_m$ describes the magnitude of the diurnal impact of light stress on PSII. Reductions in $F_v/F_m$ with respect to the previous day imply photodamage accumulation, while increases imply PSII recovery. Therefore, $F_v/F_m$ was used to assess the accumulation of PSII photo-inactivation in the symbiotic algae, *in hospite*, due to photodamage.

**Oxygen evolution**. Oxygen evolution determinations were conducted at day 0 and after 10 days of exposure to experimental conditions ($n = 4$ replicates per treatment). Corals were placed into transparent acrylic, water-jacketed laboratory-made chambers equipped with Clark-type $O_2$ electrodes (Hansatech, UK). Chambers were filled with filtered seawater from the corresponding treatments. Water motion within each chamber was provided by magnetic stir-bars. The temperature within the chambers was kept constant during the physiological determinations at the corresponding experimental treatment using a circulating bath with a controlled temperature system (RTE-100/RTE 101LP; Neslab Instruments Inc., Portsmouth, NH, USA). Light was provided by halogen lamps placed above the chambers at a particular distance to achieve an illumination of 500 µmol quanta$^{-2}$ m$^{-2}$. This irradiance allowed determination of the maximum photosynthetic rate, $P_{max}$. Irradiance levels were measured using the light sensor of a DIVING PAM (Walz GmbH, Effeltrich, Germany) previously calibrated against a cosine-corrected sensor LI-193 quantum sensor (LI-COR, Lincoln, USA). The $O_2$ electrodes were calibrated using air-saturated (100%) and nitrogen-bubbled seawater (0%) at the corresponding temperature and pH. Oxygen tension within the chambers was maintained between 20 and 80%. The rates of photosynthesis and respiration were measured continuously. As coral respiration rates were determined immediately after the organisms were exposed to $P_{max}$, we called this descriptor light-enhanced respiration ($R_L$). This parameter is a better descriptor of coral respiration, as oxygen and/or substrate availability does not limit mitochondrial activity[36]. Data were captured with a computer equipped with an A/D converter. Evolution/consumption rates within the chamber were determined from the slopes after transforming the electronic values into $O_2$ concentrations and correcting for the electrode drift [for more details see Cayabyab and Enríquez[72]; Scheufen et al.[5]]. Gross photosynthesis ($P_{max}$) was calculated by adding light-enhanced respiration to the net photosynthetic rates ($NP_{max}$). Coral P:R ratios were also determined using $P_{max}$ and $R_L$ values.

**Coral calcification**. Calcification rates were measured using the alkalinity anomaly principle based on the ratio of two equivalents of total alkalinity for each mole of precipitated calcium carbonate ($CaCO_3$)[73]. Experimental organisms ($n = 4$ replicates per treatment) were incubated for 1 hour, in 0.45 µm filtered seawater at the corresponding experimental temperature and pH. We used halogen lamps for the incubations to provide a saturating irradiance level of 500 µmol photons $m^{-2} s^{-1}$. The temperature was maintained constant in the incubation chambers using a circulating water bath with a control temperature system (RTE-100/RTE101LP; Neslab Instruments Inc., Portsmouth, NH, USA). Magnetic stir-bars allowed sufficient water motion during the incubations. Alkalinity measurements were performed immediately after the end of the incubation using a modified spectrophotometric procedure as described by Yao and Byrne[74] and Colombo-Pallotta et al.[36]. Briefly, the microtitration of each sample ($n = 3–5$) with 0.3 N HCL was performed at a flow rate of 8 µl $min^{-1}$ using a gas-tight glass syringe (Hamilton Company, Reno, USA) fitted to a syringe pump (Kd Scientific Inc., Holliston, USA) into a 1 cm length cuvette. After gently bubbling the sample with $N_2$ for at least 5 min, bromcresol green (BCG; Sigma-Aldrich, Steinheim, Germany) was added and changes in light absorbance at 444, 616, and 750 nm were monitored with an Ocean Optics USB4000 spectrophotometer (Ocean Optics, Dunedin, USA) and using a xenon light source (PX2; Ocean Optics, Dunedin, USA). At the end of the titration, the temperature was recorded for each sample using a digital thermometer with high resolution ($\pm 0.1\,°C$).

**Optical descriptions**. For reflectance (R) determinations, corals were submerged in seawater in a black container and illuminated with diffuse light reflected from a semi-spherical integrating sphere placed above the organism following a previous description[13,75]. Illumination was provided by a white LED ring placed around the coral, halogen lamps and blue and purple LEDs to enrich the spectral quality of this source. Light reflected from the coral surface was collected using a fiber-optic waveguide placed at a 45° with respect to the coral surface and at a distance of 1 cm. Reflectance spectra between 400 and 750 nm were determined on a daily basis for each experimental organism at night-time (after $F_v/F_m$ measurements) using a miniature spectrometer controlled by the manufacturer's software (USB2000+ and Spectrasuite, Ocean Optics, USA). Coral absorptance (A) was calculated as follows: A = 1 – Reflectance (R), according to Enríquez et al.[49] and Scheufen et al.[13], as light transmission through the coral skeleton can be neglected[75]. After completing the reflectance measurements, samples were frozen and stored at $-70\,°C$ until subsequent pigment and symbiont density analyses.

**Pigment content and symbiont density**. For the determination of coral pigment content and cell density, coral tissues were stripped from the skeleton by air-brushing in 0.45 µm filtered seawater (FSW). Tissue slurry was homogenized with an Ultra-Turrax® T10 Basic homogenizer (IKA®, Staufen, Germany) and the homogenate was centrifuged at 2000 rpm for 10 min. Tissue protein content was spectrophotometrically determined on the supernatant based on the differential absorbance between 235 and 280 nm[76]. The remaining algal pellet was resuspended in FSW, and three subsamples were taken for analysis of photosynthetic pigment concentration. *Symbiodininaceae* pigment extraction was performed

using acetone:dimethyl sulfoxide (20:1 v/v)[77], and pigment concentration was determined spectroscopically using the equations of Jeffrey and Humphrey[78]. In addition, aliquots were fixed with Lugol iodine solution for symbiont cell counts, which were performed using an Improved Neubauer hematocytometer ($n = 8$ replicate counts).

**Statistics and reproducibility.** Two-way analyses of variance (ANOVA) and Tukey HSD post hoc tests were used to determine significant effects of each experimental treatment (elevated temperature, low pH and their combined effect) on coral physiology and pigmentation and to identify significant differences between treatments, respectively. Welch t-test were used to assess the difference in $F_v/F_m$ between successive days. Assumptions of normality and homogeneity of variances were tested using a Shapiro-Wilks test and a Levene test, respectively. If required, data were subjected to transformations to obtain normal distributions and homogenous variances. All data were analyzed in SPSS Statistics 20.0 (IBM Inc., USA).

**Reporting summary.** Further information on research design is available in the Nature Portfolio Reporting Summary linked to this article.

## Data availability

All data are available in the main text, in the supplementary materials and in the Supplementary Data 1 document.

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

## Acknowledgements
We would like to thank all members of the laboratory of photobiology of the UASA-UNAM, particularly Román M. Vásquez-Elizondo, for assistance during the experimental work and fruitful discussions. The European Union is also acknowledged for the grant support 244161, EU-PFP7 to S.E. and R.I.-P.; as well as the CONACYT from Mexico for the grant 129880 to S.E. (Conv-CB-2009). Finally, we also thank the DGAPA-UNAM for the financial support of a postdoctoral-fellowship to W.E.K. and for the support of a sabbatical period at PSU to S.E. with a PASPA fellowship during the final writing of the manuscript.

## Author contributions
Conceptualization: S.E. and R.I.-P. (designed research). Investigation: W.E.K. (performed the experiment). Analyses: W.E.K. and S.E. Supervision: S.E. Writing—original draft: W.E.K. and S.E. Writing—review & editing: W.E.K. and S.E.

## Competing interests
The authors declare no competing interests.
