## [Peer Review File · Communications Biology]

Reviewers' comments:

Reviewer #2 (Remarks to the Author):

This manuscript by Krämer et al. uses results from previous studies and the findings from an experiment to address gaps in our understanding of the physiological impacts of ocean acidification on corals.

The authors' reasoning that few ocean acidification studies measure photosynthesis and calcification is a strong justification and highlights the importance and relevance of their study.

Overall, I think this is an interesting study. I appreciate that the team quantified many of the parameters that should be measured in these types of studies. However, the main finding support what is known and expected from similar studies, warming induces a more adverse effect on coral physiology than ocean acidification.

Nevertheless, I think this is a worthwhile study. Although, I wonder if the experiment ran for more than ten days if the ocean acidification results would become more interesting and informative. Also, throughout the manuscript, the multiple shifts in text font were a distraction, and some figures were improperly referenced.

The presentation of the findings in the abstract is uninteresting, and I think this can be significantly improved. In fact, after reading the abstract, I was not sure what the study's main contribution was.

Understanding the figures require a lot of back and forth from figure to figure legend. Figure 2 is nice, but figure 1 and 3 could be improved. If allowed, consider using the species name in the figure panels for clarity. I think this is why figure 2 works. There is too much going on in figure 3. I am assuming this is due to the number of figures allowed by the journal, but this should not sacrifice the clarity of the figures.

I did not find any concerns with the methods. These are all standard in coral studies, and it was easy to follow what the authors did in each case. I commend them for measuring all these time-consuming parameters. The statistics seem appropriate, and I think the work is easily reproducible. I hope to see a study with an experiment running much longer than ten days.

I would have liked a more in-depth analysis of the previous studies. I think the finding in the current study could have been better tied to previous work.

Overall, I think this is a valuable study and aspects of the findings are interesting, but I think there is room for improvements, as I outlined above in my comments.

Reviewer #3 (Remarks to the Author):

The manuscript reports on the combined effects of elevated temperature stress and OA on the physiology of four coral species. The results show that the effect of heat stress is more important than OA. The two way anova largely reports this conclusion but posthoc tests are inappropriate and should be revised. The experimental design is not perfect (replication may be a problem) but it is clearly stated, adding some indication on turn over of water in the two experimental tanks and the common header tanks could help in decreasing the "pseudo-replication" suspicion.

The discussion should be revised as some of the conclusion are not supported by the results. For example the authors state that *Orbicella faveolata* presents one of the largest adverse impacts of OA

on calcification, while the results show that OA do not have any effect. The review of the different studies is interesting and could be further developed in the discussion as for the moment it is quite limited. Mostly the only conclusion is on the number of studies that measured certain parameters.

This manuscript should therefore be substantially revised if it is to be published.

Detailed comments:

Intro

line 73: "Not such a tipping point has been detected at this temperature for coral respiration", If my understanding is correct, I agree but I feel the sentence is a little out of place and confusing. Perhaps stating that this threshold temperature is lower than the ABT for respiration, or that such a threshold has not yet been defined for respiration, could help clarify.

line 93: I feel the use of "very few" is a little harsh, there are many reports (not enough I agree) on the effect OA+Warming on photosynthesis and calcification.

line 100: "Assuming that carbon availability limits both coral calcification and photosynthesis" this is a big assumption, and many studies, including the lack of increase of photosynthesis on OA, suggest that corals are not carbon limited as they are able to use carbon concentration mechanisms. But perhaps the authors wants to say that under the hypothesis that OA would increase photosynthesis was made under this assumption?

line 110: I am not sure what "this effect" is referring to? Bleaching?

line 111: I don't understand what the relation between "light can modulate the susceptibility of coral calcification to OA" and "the high sensitivity of corals to light stress under elevated temperature" ? High sensitivity to light under heat stress is related to the production of ROS, which to my knowledge has not been linked to OA or (directly) to calcification.

Results

The author should indicate the results of the 2 way ANOVA in a clear way (stating the F, df, and p-value in the text would be nice or at least similar than what is shown line 168) and not directly indicate the results of the posthoc (inappropriate t-tests (!)). Moreover the grouping of coral species may indicates that the authors want to reach a multi species conclusion, so perhaps they could consider mixed models with species as a random effect?

Most of the results could be summarized as mostly only the increase in temperature affected the corals and symbionts physiology (describe the main effects of this treatment comparing temperature effect regardless of OA). The only exceptions are: Chlo a and P:R in *O. annularis* and Gmax in *P. strigosa* for which a significant effect of pH was found (in this case as there is no interaction but two significant main effect, the direction (increase or decrease) under elevated pH combining temperature levels should be described. Then, Fv/Fm and Pmax in *P. strigosa* show an interaction of temperature and pH and therefore the interactions needs to be describe with posthoc.

line 134: How was the "significant recovery" of Fv/Fm statistically tested? This is not indicated in the method. Considering these are repeated measurement, a 2way anova cannot be used and mixed model should be preferred. Moreover if the authors also used multiple t-test to compare to the previous day, with any adjustment of the alpha, there is a strong chance of type I error.

line 140: The statements "Changes were also insignificant for the OA treatment," and the following

description of the results showing significant changes are in contradiction. Perhaps changing to "Changes were mostly insignificant or of only minor amplitude for the OA treatment, with only *P. strigosa* [...]"

line 155: It is slightly confusing to repeat the result for respiration in the OA treatment here.

line 157: Because of the order of the manuscript, it is the first time that CT-COA and CT-OA are referred to. The choice of using COA for "control ocean acidification" is confusing in my opinion. Perhaps CCO₂ would be better as OA refers to the increase of CO₂. However here it is perhaps not needed to use the abbreviations and I would suggest to emphasize that these observations are valid for the control temperature. A way to phrase it could be: "The ratio of P_{max} to RL (P/R) did not change with levels of CO₂ under control temperature level during the experiment."

line 164: Why is "values not different from zero" stated? Were Net calcification rates equal to zero omitted? If yes why? This would indicate a strong decrease of calcification

line 171: *M. cavernosa* G_{max} is not affected by pH, so this statement is not valid.

line 174: the effect of OA on P_{max} in *O. faveolata* is $F = 0.005$, $P = 0.944$ and there is clearly no effect of OA on P_{max} from Fig 2 panel b... Same for symbiont density

line 180: It is strange to report the results of another study in the result section. This should be limited to discussion.

Discussion

line 191: Rephrase: " A thermal anomaly of +2°C caused in then days severe disturbances "

line 213: By coral pigmentation, I assume the authors mean the zooxanthellae chlorophyll a contents. A strong decrease of chl *a* or Fv/Fm are a suitable proxy for coral bleaching as they are a definition of coral bleaching. The second half of the sentence is correct, these parameters may not actually quantify the stress impact on corals as they do not always translate in a decrease in photosynthesis rates. However, decline photosynthesis may also not be suitable as a "single" indicator for stress as increased production of ROS for example can lead to apoptosis which is a very strong stress. I think what the author highlight here, is the importance of using multiple parameters in assessing the impact of heat stress.

line 218: Why is it stated that "*O. annularis* and *M. cavernosa*, were the more sensitive to light-stress". The effect of light was not tested, and a decline of Fv/Fm was observed only under heat stress, without a comparison with heat stress under lower light level, it is not possible to conclude on the sensitivity of light. Or is it s reference?

line 225: "Our results, then, do not support the conclusion that light has the capacity to modulate the susceptibility of coral calcification to acidified conditions³⁹, or that OA is a key factor in the induction of coral bleaching²², although they support the importance to discern between the dysfunctional ("bleached") and the "stressed" coral phenotypes¹³". This sentence needs to be reviewed. The three statements are unrelated and do not link with the previous sentence. Moreover, as light was not tested, how can the author conclude on the effect of light under OA? The second statement, OA does not induce bleaching is true (although only one study showed this and it is not commonly accepted), and the third statement is true but unrelated to OA.

line 241: While I agree that respiration measurement should be included in coral bleaching research, it would be nice to indicate in clear term how this study does indicates its importance as only 2 species showed a significant effect of temperature on respiration and none showed an effect of pH.

line 245: " We observed increases in coral respiration at enhanced DIC supply (Dissolved Inorganic Carbon, OA-treatment) in the four species, consistent with previous findings and with a transcriptomic analysis" I am confused, no effect of pH was found. The only "increased" in respiration was found for *P. strigosa* using posthoc but as the main effect of pH was not significant and there is no interaction, this posthoc test may not be suitable. Moreover once corrected for multiple comparison, it may not be significant anymore. But regardless, the three other species does not show any effect of OA on respiration.

line 262: See my previous comment on this grouping. the effect on *M. cavernosa* is not significant I think. So how does the author justify the grouping? Same for *O. faveolata*, I do not see any significant of OA on symbiont related parameters, so why is it grouped with *P. strigosa*?

line 286: The fact that the experimental study here does not support the metaanalysis results is not a problem, this is why metanalysis are done, to bring conclusion over several species, experimental design and studies.

line 289: Yes experimental analysis are required, and then can be combined within metanalysis to bring about general conclusion. These are not exclusive contrary to what it reads like here.

line 292: "the strength of the model system under analysis." i am not sure what is a "model system".

line 298: see previous comment on light-stress

line 305: see my previous comment on the grouping of the species, and therefore this conclusion on two different pathways for carbon allocation. I think it is not supported here.

line 309: "*Orbicella faveolata*, also presents one of the largest adverse impacts on calcification" : but all measurements on *O. faveolata* show that there was no significant effect of OA...

Methods

line 332: The indications on how the carbonate chemistry was measured (and calculated) could be moved here.

Experimental design: the author used only two tanks per treatment with 2 nubbins per species per tank. Many would consider that the true replication in this case is $n = 2$, but a recent guidelines stated that at least 2 tanks should be used for short term experiments (<7 days)... Moreover, in this study the temperature in the 2 tanks per treatment was regulated in a common header tank, same for OA. So the 2 tanks are not truly independent. In such case it could be concluded that the experiment is pseudoreplicated... Could the author provide an idea of what was the turnover of water in the header tank and the experimental tanks? As it is running water, it is possible that a high turnover would negate any potential tank effects.

line 383: Indicate how was the difference among days tested. If multiple t-tests were used, the alpha should be adjusted for multiple comparison.

Statistical analysis need to be revised. Except for a few case, the effect of pCO₂ is not significant in the 2 way anova, and only twice an interaction of pH and temperature is showed. Under this conditions, it is not advisable to conduct posthoc tests among individual cross treatments (except in the case of interaction). Moreover the authors used individual t tests for multiple comparison which is really not appropriate (type I error). The author must revise the statistic and use more appropriate posthoc tests (Tukey HSD or Dunnet if the author only want to compare to control). Some data seems

to have been transformed to validate the ANOVA assumptions but the transformations are not given, if it is too long at least these could be included in the supplementary table.

Figures

Figure 3: The panel a-h shows the same results but in a more complicated way than figure 2 in my opinion. Panel c and d are somewhat interesting but not really discussed in the manuscript. Panels i to l show results from a different study, and should be removed in my opinion as it only confuses the reader who is trying to understand the link with the current study.

Tables

Table 2: In addition to the fact that multiple t-tests cannot be used. Why is the degree of freedom different. It should be 6 but in some times, 5, 4 or even 2.17 ? This would indicate that some replicates were removed from the analysis and should be justified.

Response to the reviewer's comments in blue:

Reviewer #2 (Remarks to the Author):

This manuscript by Krämer et al. uses results from previous studies and the findings from an experiment to address gaps in our understanding of the physiological impacts of ocean acidification on corals.

The authors' reasoning that few ocean acidification studies measure photosynthesis and calcification is a strong justification and highlights the importance and relevance of their study.

Overall, I think this is an interesting study. I appreciate that the team quantified many of the parameters that should be measured in these types of studies. However, the main finding support what is known and expected from similar studies, warming induces a more adverse effect on coral physiology than ocean acidification.

We would like to thank the reviewer for his/her positive comments about our study. However, we don't fully agree with the statement that our main conclusion is already known and expected. For coral physiologists this finding could be nowadays more evident, although almost impossible to defend and publish a few years ago, due to the large variability and inconsistent results documented in the literature for the OA response. However, what it is perhaps more important is the lack of incorporation of this information in coral ecology, where both global threats are still considered of similar relevance. Our aim in the present study was not only to evidence the moderate effects of OA relative to the impact of global warming, but to emphasize the still incomplete characterization of the OA response of symbiotic corals, thanks to the combination of an experimental analysis and a literature review. This conclusion may contribute to stimulate future physiological studies focused on elucidating the biological mechanisms involved in this response. Another potential contribution of our study is to counterbalance the idea that experimental approaches cannot elucidate this response, due to the "impossibility to simulate in the laboratory the multiple natural interactions that may explain this response". We do believe that the "species-specific" component of the large variability described in the literature can be understood with a correct identification of the main targets and/or processes (i.e., biological mechanisms) impacted by climate change stressors.

Nevertheless, I think this is a worthwhile study. Although, I wonder if the experiment ran for more than ten days if the ocean acidification results would become more interesting and informative. Also, throughout the manuscript, the multiple shifts in text font were a distraction, and some figures were improperly referenced.

We apologize for the text problems of the previous manuscript. Something was wrong in the document transfer, but hope they have been resolved in the revised version. The reviewer is right that a longer experiment may have allowed quantifying the magnitude of the impact of low pH on the physiology of symbiotic corals, or simply finding that there is no effect for many species at the physiological level although the costs of living may increase. However, these conclusions were not considered in the objectives of our analysis. Indeed, we don't conclude that ocean acidification has no effect on coral physiology, but that this effect is less severe than that induced by thermal stress, and 10 experimental days were sufficient to demonstrate this.

The presentation of the findings in the abstract is uninteresting, and I think this can be significantly improved. In fact, after reading the abstract, I was not sure what the study's main contribution was.

We have changed the abstract to resolve the problem indicated by the reviewer. However, due to text restrictions (150 words), we decided to maintain the emphasis on the low consensus about the impact of ocean acidification on coral physiology despite three decades of coral research, in

comparison with the well-known adverse effects of thermal stress. In addition, we wanted to stress that this uncertainty is in part derived from partial views of this response due to incomplete experimental characterizations. Another important conclusion we wanted to emphasize in the abstract, critical to predict the future of coral reefs, is that the large species-specific component of this moderate OA response has to be understood with “a correct identification of the main targets and/or processes impacted by climate change stressors”. The new abstract is as following:

After three decades of coral research on the impacts of climate change, there is a wide consensus on the adverse effects of heat-stress, but the impacts of ocean acidification (OA) are not well established. Using a review of published studies and an experimental analysis, we confirm the large species-specific component of the OA response, which predicts moderate impacts on coral physiology and pigmentation by 2100 (scenario-B1 or SSP2-4.5), in contrast with the severe disturbances induced by only +2°C of thermal anomaly. Accordingly, global warming represents a greater threat for coral calcification than OA. The incomplete understanding of the moderate OA response relies on insufficient attention to key regulatory processes of these symbioses, particularly the metabolic dependence of coral calcification on algal photosynthesis and host respiration. Our capacity to predict the future of coral reefs depends on a correct identification of the main targets and/or processes impacted by climate change stressors.

Understanding the figures require a lot of back and forth from figure to figure legend. Figure 2 is nice, but figure 1 and 3 could be improved. If allowed, consider using the species name in the figure panels for clarity. I think this is why figure 2 works. There is too much going on in figure 3. I am assuming this is due to the number of figures allowed by the journal, but this should not sacrifice the clarity of the figures.

We have changed the figures according to reviewer's comments to make them easier to follow. Figure 1 has now the names of the species on the top of each panel. Figure 3 has been simplified removing the analysis from Scheufen et al. (2017) data. We add a new figure 4 to illustrate the new PCA analysis performed and the average values for each of the two groups identified.

I did not find any concerns with the methods. These are all standard in coral studies, and it was easy to follow what the authors did in each case. I commend them for measuring all these time-consuming parameters. The statistics seem appropriate, and I think the work is easily reproducible. I hope to see a study with an experiment running much longer than ten days.

We thank the reviewer for his/her comments. Considering the complexity of this experimental study and the fact that the main objective was to compare the direct and combined responses of four different species to four experimental treatments, we found 10 experimental days sufficient to characterize these differences. A longer experiment would have been unaffordable for the magnitude of this comparison, which also used 9-10 parameters for the characterizations. Indeed, the longer is the maintenance of the corals in experimental tanks the more likely is the potential interference of other adverse effects such as, for example, coral starving.

I would have liked a more in-depth analysis of the previous studies. I think the finding in the current study could have been better tied to previous work.

We also agree with the reviewer that it could have been more interesting an in-depth analysis of previous studies. Unfortunately, we found this in-depth analyses impossible to perform considering that most of the studies published present incomplete characterizations.

Overall, I think this is a valuable study and aspects of the findings are interesting, but I think there is room for improvements, as I outlined above in my comments.

We thank the reviewer for his/her comments and hope the revised version has incorporated most of the improvements suggested.

Reviewer #3 (Remarks to the Author):

The manuscript reports on the combined effects of elevated temperature stress and OA on the physiology of four coral species. The results show that the effect of heat stress is more important than OA. The two way anova largely reports this conclusion but posthoc tests are inappropriate and should be revised. The experimental design is not perfect (replication may be a problem) but it is clearly stated, adding some indication **on turn over of water in the two experimental tanks and the common header tanks could help in decreasing the "pseudo-replication" suspicion.**

The discussion should be revised as some of the conclusion are not supported by the results. For example the authors state that *Orbicella faveolata* presents one of the largest adverse impacts of OA on calcification, while the results show that OA do not have any effect. The review of the different studies is interesting and could be further developed in the discussion as for the moment it is quite limited. Mostly the only conclusion is on the number of studies that measured certain parameters.

This manuscript should therefore be substantially revised if it is to be published.

We thank the reviewer for the positive comments and helpful suggestions. We hope we could resolve satisfactorily the main problems indicated. The previous posthoc tests have been removed from the manuscript. The new Table 2 show Tukey HSD tests in support of the differences found in the coral response to the experimental treatments. We would like also to clarify that we did not conclude in our study that *Orbicella faveolata* presents one of the largest adverse impacts of OA on calcification, as the OA impact was found moderate for all species, including that of *O. faveolata*. The reviewer may have misinterpreted the final sentence of the discussion: "*only a better knowledge of this regulation may allow elucidating the impact of climate change on symbiotic corals, and perhaps explain why a robust species to heat-stress such as **Orbicella faveolata**, also presents one of the largest adverse impacts on calcification.*" We found this "apparent" paradox very relevant to stress the importance to understand why a robust photosynthetic response to heat stress was found also associated with the largest reductions/impact in coral calcification (i.e., *O. faveolata* was able to maintain a functional photosynthesis after exposure to heat stress, showing the lowest declines in P_{max} among the four species investigated, as well as one of the largest reductions in calcification). The ecological implications of this type of response are important, as the selection of robust species to global warming may not be the solution for the survival of coral reefs if coral calcification and in consequence, reef accretion, is also compromised by heat stress. We have changed the former sentence to: "*also presents one of the largest adverse impacts of **global warming** on calcification*", in order to better clarify the previous sentence.

Detailed comments:

Intro

line 73: "Not such a tipping point has been detected at this temperature for coral respiration", If my understanding is correct, I agree but I feel the sentence is a little out of place and confusing. Perhaps stating that this threshold temperature is lower than the ABT for respiration, or that such a threshold has not yet been defined for respiration, could help clarify.

Text corrected following the reviewer indications (current lines 71-72)

line 93: I feel the use of "very few" is a little harsh, there are many reports (not enough I agree) on the effect OA+Warming on photosynthesis and calcification.

"Very few" has been changed by "a limited number of studies" (current line 93).

line 100: "Assuming that carbon availability limits both coral calcification and photosynthesis" this is a big assumption, and many studies, including the lack of increase of photosynthesis on OA, suggest that corals are not carbon limited as they are able to use carbon concentration mechanisms. But perhaps the authors wants to say that under the hypothesis that OA would increase photosynthesis was made under this assumption?

Yes. We just wanted to say that the hypothesis that OA will increase coral photosynthesis is based on this assumption, but, we agree with the reviewer, that there is evidence that does not support it, and, thus, that this assumption is not correct. To prevent misunderstandings, we changed the sentence as following: "*It has been hypothesized that OA may increase coral photosynthesis rates by inducing moderate enhancements in pCO₂, assuming that carbon availability limits both coral calcification and photosynthesis, a controversial assumption not yet demonstrated. While slight increases in net photosynthesis have been reported, most studies have concluded that coral photosynthesis is in general unaffected by OA*". (current lines 98-102)

line 110: I am not sure what "this effect" is referring to? Bleaching?

We refer here to the adverse effects of OA on coral physiology. The text has been changed, removing "*this effect*" by "*the OA adverse effects*". (current line 110)

line 111: I don't understand what the relation between "light can modulate the susceptibility of coral calcification to OA" and "the high sensitivity of corals to light stress under elevated temperature" ?

High sensitivity to light under heat stress is related to the production of ROS, which to my knowledge has not been linked to OA or (directly) to calcification.

Suggett et al. (2013) in the paper entitled: *Light availability determines susceptibility of reef building corals to ocean acidification*, concluded exactly this. Our results don't support this conclusion, but we considered important to mention this previous interpretation of the OA response of symbiotic corals in the introduction of our study. We don't think that this conclusion is inappropriate, as under light stress, the costs of maintenance of the photosynthetic apparatus of the symbionts increase significantly, as well as the accumulation of photodamage with the consequent reduction in photosynthesis (i.e., photoinhibition). Thus, light stress not only induces oxidative stress, but it may also reduce the metabolic energy translocated from algal photosynthesis to the host in support of coral calcification. Any significant adverse effect of OA on coral photosynthesis could also significantly impact on coral calcification. The lack of support of this interpretation by our study is derived from the fact that we did not observe any effect of OA on algal photosynthesis, in accordance with a previous analysis performed also in our laboratory for coralline algae (Vásquez-Elizondo and Enríquez 2016).

Results

The author should indicate the results of the 2 way ANOVA in a clear way (stating the F, df, and p-value in the text would be nice or at least similar than what is shown line 168) and not directly indicate the results of the posthoc (inappropriate t-tests (!)). Moreover the grouping of coral species may indicates that the authors want to reach a multi species conclusion, so perhaps they could consider mixed models with species as a random effect?

Thank you for the suggestion. Due to text restrictions, unfortunately, we cannot include the ANOVA statistics in the text. If the editors open us the option to include this information, we will be pleased to do it. As indicated previously, the posthoc tests of the former Table 2 have been replaced by a new table 2 with Tukey HSD tests in support of the differences observed in the coral responses. The

figures have been also corrected according to the new results. Additionally, in the revised manuscript we have performed a PCA Analysis to better illustrate the multi species variability, using the four samples characterized for each species. The former two groups identified are confirmed in this new multivariate analysis, which also highlights the parameters that better define the differences found between groups for the OA response. The PCA analysis revealed that the differences among species were not so well defined due to *O. faveolata*. Two samples of this species belonged to one group whereas one sample fit into the other group. The fourth replicate characterized for this species showed low physiological performance, and should have been excluded from the analysis, although we decided to maintain it in the PCA. The variability observed for *O. faveolata* indicates that four replicates were not enough to characterize the large variation displayed by this species.

Most of the results could be summarized as mostly only the increase in temperature affected the corals and symbionts physiology (describe the main effects of this treatment comparing temperature effect regardless of OA). The only exceptions are: Chlo a and P:R in *O. annularis* and Gmax in *P. strigosa* for which a significant effect of pH was found (in this case as there is no interaction but two significant main effect, the direction (increase or decrease) under elevated pH combining temperature levels should be described. Then, Fv/Fm and Pmax in *P. strigosa* show an interaction of temperature and pH and therefore the interactions needs to be describe with posthoc.

Thank you for these suggestions that have allowed us to simplify the text in the results section.

line 134: How was the "significant recovery" of Fv/Fm statistically tested? This is not indicated in the method. Considering these are repeated measurement, a 2way anova cannot be used and mixed model should be preferred. Moreover if the authors also used multiple t-test to compare to the previous day, with any adjustment of the alpha, there is a strong chance of type I error.

Fv/Fm is a parameter that presents large variability among days and a strong dependence on the variation of solar irradiance. The origin of this variability is well-known in plant physiology as responds to the capacity of repair of the photosynthetic apparatus of the species in relation to the rate of photodamage accumulation during a particular "working" day or period. This variability is present in control organisms, as it depends on daily changes in irradiance. Overcast days can be beneficial for all photosynthetic organisms, as they allow repairing some of the damage accumulated in previous days, allowing enhancing Fv/Fm values in relation to previous days. What we found interesting to emphasize from our results, is the higher capacity of repair of *O. annularis* during overcast days in comparison with the absence of such repair in *M cavernosa* (i.e., no enhancement in Fv/Fm). To test this significance, we compared Fv/Fm values during overcast days in relation to the values recorded the day before to this period of lower irradiance.

line 140: The statements "Changes were also insignificant for the OA treatment," and the following description of the results showing significant changes are in contradiction. Perhaps changing to "Changes were mostly insignificant or of only minor amplitude for the OA treatment, with only *P. strigosa* [...]"

Text was corrected according to reviewer's suggestions (see line 140).

line 155: It is slightly confusing to repeat the result for respiration in the OA treatment here.

Thanks. We have removed this sentence from the text, as the reviewer is correct that the OA results were described previously and this information was superfluous and confusing in this paragraph.

line 157: Because of the order of the manuscript, it is the first time that CT-COA and CT-OA are referred to. The choice of using COA for "control ocean acidification" is confusing in my opinion. Perhaps CCO₂ would be better as OA refers to the increase of CO₂. However here it is perhaps not needed to use the abbreviations and I would suggest to emphasis that these observation are valid

for the control temperature. A way to phrase it could be: "The ratio of Pmax to RL (P/R) did not change with levels of CO2 under control temperature level during the experiment."

Thank you, we corrected the text according to the reviewer's suggestion.

line 164: Why is "values not different from zero" stated? Were Net calcification rates equal to zero omitted? If yes why? This would indicate a strong decrease of calcification

We determined the instantaneous calcification rate for four replicates under the same treatment. When the average for the four samples did not allow estimating a positive or negative value, and thus the estimation was not different from zero considering the STD of this determination, we assumed that calcification was fully suppressed but that decalcification was not observed. Positive and negative values required the estimation of significant differences with respect to 0.

line 171: M cavernosa Gmax is not affected by pH, so this statement is not valid.

line 174: the effect of OA on Pmax in O faveolata is $F = 0.005$, $P = 0.944$ and there is clearly no effect of OA on Pmax from Figure 2 panel b... Same for symbiont density

The reviewer is correct in both comments. In the revised manuscript, we performed a PCA analysis to better compare and identify the differences and similarities among species in the OA response. Two clusters were identified for all the coral samples. One cluster was formed by *O. annularis* and *M. cavernosa*, as well one sample of *O. faveolata*. The other cluster was formed by all samples of *P. strigosa* and two of the samples of *O. faveolata*. We think this analysis allowed a better identification of the same two groups documented previously and despite the non-significant differences observed among species due to the large variability displayed by *O. faveolata*. We truly appreciate the reviewer's comment, as it forced us to find a better way to illustrate these differences.

line 180: It is strange to report the results of another study in the result section. This should be limited to discussion.

We removed this paragraph from the revised manuscript.

Discussion

line 191: Rephrase: " A thermal anomaly of +2°C caused in then days severe disturbances "
Sentence corrected as following: "*Applying a thermal anomaly of +2°C for 10 days, we observed severe disturbances on coral photosynthesis and calcification, whereas the experimental simulation of the expected OA conditions by 2100...*" (current lines 194-196).

line 213: By coral pigmentation, I assume the authors mean the zooxanthellae chlorophyll a contents. No. Coral pigmentation does not only depend on the variation of symbiont cell pigmentation (zooxanthellae chl a content, Ci) but also on the variation in the number of symbionts. Both parameters regulate the variation in coral pigmentation (see Scheufen et al. 2017b).

A strong decrease of chl a or Fv/Fm are a suitable proxy for coral bleaching as they are a definition of coral bleaching. Unfortunately, chl a and Fv/Fm are not suitable proxies for coral bleaching, although the reviewer is right that these parameters are commonly used for the identification of coral bleaching. We don't agree with the definition of coral bleaching limited to the detection of severe reductions in coral pigmentation and symbiont content, as it is not enough to detect a dysfunctional condition of the symbiotic relationship (see Scheufen et al. 2017a and b). In these previous studies, we documented this problem, and we found an efficient photoacclimatory response to heat (30°C) in spring for *O. annularis*, characterized by significant declines in coral pigmentation and symbionts but no changes in the initial photosynthetic rates. Accordingly, it was not correct to interpret that this

temperature induced severe stress looking, only, at the large reductions in pigmentation, because photosynthesis was not altered. The interpretation that these “paled organisms” of *O. annularis* were bleached, was even more incorrect. All these findings led us to conclude that the “dysfunctional” condition of a symbiotic coral defined as “coral bleaching” requires a better physiological identification, and postulate that the physiological condition where algal photosynthesis is fully suppressed, could be a better proxy. We are aware of the unfeasibility of using this identification of bleaching in ecological surveys, which need to rely on simple parameters. However, this limitation cannot constrain the physiological understanding of coral bleaching and the searching for more accurate identifications for this important dysfunctional condition, which defines a disruption in the symbiotic association. Moreover, a correct identification of coral bleaching is fundamental for transcriptomic, genomic and/or proteomic approaches. Another point that may need to be clarified, is that before the bleaching phenotype is achieved, the “stressed coral” has the capacity to prevent bleaching using different mechanisms. Recently, Gómez-Campo et al. (2022) has presented a model that aims to explain the complexity of homeostatic processes that can be induced under stress before the bleached phenotype is developed. This condition, thus, has to be considered to occur only at the end of the stress period, when all homeostatic processes are overwhelmed.

The second half of the sentence is correct, these parameters may not actually quantify the stress impact on corals as they do not always translate in a decrease in photosynthesis rates. However, decline photosynthesis may also not be suitable as a "single" indicator for stress as increased production of ROS for example can lead to apoptosis which is a very strong stress. I think what the author highlight here, is the importance of using multiple parameters in assessing the impact of heat stress.

We agree with the reviewer that stress quantification should require multiple parameters. In our study, we don't consider that the decline in photosynthesis is a measure of stress. In fact, we don't quantify the stress. What we propose is the use of the full suppression of coral photosynthesis for the identification of the bleached phenotype. If coral photosynthesis is still functional (i.e., it has positive values), we consider that the coral has not yet developed the bleached phenotype and thus that the symbiosis is not yet disrupted, as the remaining symbionts are still performing photosynthesis. However, when photosynthesis is fully suppressed, the bleached phenotype is for sure achieved, independently of coral pigmentation (we have observed this phenotype before the pale condition was achieved, for example, in *M. cavernosa*).

line 218: Why is it stated that "*O. annularis* and *M. cavernosa*, were the more sensitive to light-stress". The effect of light was not tested, and a decline of Fv/Fm was observed only under heat stress, without a comparison with heat stress under lower light level, it is not possible to conclude on the sensitivity of light. Or is it s reference?

The fluorescence technique allows us to monitor differences among species in their capacity to use solar energy to fix organic carbon under different treatments, minimizing damage on their photosynthetic apparatus and the costs of repair it. These differences are monitored using the variation in Fv/Fm under similar light fields, which allows identifying the more sensitive species to light stress, as they accumulate more photodamage in their membranes despite they were exposed to similar experimental light conditions. Accordingly, the availability of this powerful physiological tool allows the characterization of these differences, without the need to use different light treatments. However, the experiment requires a strong control of the light field, minimizing the variability within and among tanks. In the present study, we observed that the symbionts of *O. annularis* and *M. cavernosa* showed larger accumulation of photodamage (larger reductions in Fv/Fm) than those of the other two species, which supports the higher sensitivity of these symbionts *in hospite* to light stress.

line 225: "Our results, then, do not support the conclusion that light has the capacity to modulate the

susceptibility of coral calcification to acidified conditions³⁹, or that OA is a key factor in the induction of coral bleaching²², although they support the importance to discern between the dysfunctional ("bleached") and the "stressed" coral phenotypes¹³". This sentence needs to be reviewed. The three statements are unrelated and do not link with the previous sentence. Moreover, as light was not tested, how can the author conclude on the effect of light under OA? The second statement, OA does not induce bleaching is true (although only one study showed this and it is not commonly accepted), and the third statement is true but unrelated to OA.

The three statements of this sentence refer to previous conclusions published by different studies. Suggett et al. (2019) concluded that light stress determines the sensibility of corals to ocean acidification, while Anthony et al. (2008) concluded that ocean acidification can induce coral bleaching. None of these conclusions were supported by our analysis. However, our study did support the conclusion by Scheufen et al (2017a,b), who proposed the relevance of discerning between the stressed and bleached phenotypes. We changed this sentence in order to better clarify the text, as following: "*Our results, then, do not support the conclusion that the level of light stress "sensed" by the algae in hospite, regulates the susceptibility of coral calcification to acidified conditions or that OA can induce coral bleaching. However, they support the importance of discerning between the dysfunctional ("bleached") and the "stressed" coral phenotypes previously proposed (see lines 231-235).*"

line 241: While I agree that respiration measurement should be included in coral bleaching research, it would be nice to indicate in clear term how this study does indicates its importance as only 2 species showed a significant effect of temperature on respiration and none showed an effect of pH.

The importance of the variation in holobiont respiration does not only rely on the finding of significant values for this parameter, but also on the understanding of the consequences for the ratio photosynthesis/respiration (P/R). In our study, only one species, *O. faveolata*, showed a P/R value of 1 after the heat-stress treatment. All the other P/R values estimated were below 1 (for the combined treatment of *O. faveolata* and for all heat stress treatments of the other three species). This finding is very relevant as it indicates that heat stress induced negative metabolic balances (P/R < 1) and at light saturation (P_{max}) in almost all cases investigated. We thought relevant to stress the finding of such adverse metabolic condition in the experimental corals and tried to explain it better in the text. In general and in addition to the importance of P/R, our aim is to emphasize that the understanding of the impact of any stress on coral calcification needs to pay attention to both metabolic rates, algal photosynthesis and holobiont respiration, as they are the metabolic support for coral calcification (see lines 251 to 262).

line 245: " We observed increases in coral respiration at enhanced DIC supply (Dissolved Inorganic Carbon, OA-treatment) in the four species, consistent with previous findings and with a transcriptomic analysis" I am confused, no effect of pH was found. The only "increased" in respiration was found for *P. strigosa* using posthoc but as the main effect of pH was not significant and there is no interaction, this posthoc test may not be suitable. Moreover once corrected for multiple comparison, it may not be significant anymore. But regardless, the three other species does not show any effect of OA on respiration.

The reviewer is right that the differences in coral respiration were not significant. However, we thought relevant to emphasize this observation considering that increased transcription of genes associated with respiration was documented by Davies et al. (2016) for the coral *Siderastraea siderea*. We modified the text to clarify that our study was not able to support any clear conclusion about a possible beneficial effect of OA on coral respiration, but we maintain this sentence as this observation may stimulate future work in order to confirm or reject a potential effect of OA on coral respiration (see lines 252 to 255).

line 262: See my previous comment on this grouping. the effect on *M. cavernosa* is not significant I

think. So how does the author justify the grouping? Same for *O. faveolata*, I do not see any significant of OA on symbiont related parameters, so why is it grouped with *P. strigosa*?

We agree with the reviewer that this grouping by species was not so clear due to the lack of significance observed for most parameters. In the revised manuscript, we performed a PCA analysis using the four replicates characterized for each species and found the same two groups previously described, as well as the parameters that better differentiate them. These groups, however, were not defined by species, as only three species, *M. cavernosa*, *O. annularis*, and *P. strigosa*, responded to the previous grouping. For *O. faveolata*, two samples were associated with the response of *P. strigosa*, and one sample clustered into the other group, associated with the response of *M. cavernosa* and *O. annularis*. The fourth replicate characterize for this species showed large reductions in all parameters, including calcification, indicating low performance for this particular sample. In the first group, *M. cavernosa*, *O. annularis* and one replicate of *O. faveolata*, we observed significant increases in calcification, G_{max} , and reductions in chl_a and symbiont content. For *P. strigosa* and two samples of *O. faveolata*, we observed increases in P_{max} , R, pigment and symbiont content and a slight reductions in calcification in comparison with the other group. So, we believe that the new PCA analysis allowed resolving the problem caused by the large variability displayed by *O. faveolata*.

line 286: The fact that the experimental study here does not support the metaanalysis results is not a problem, this is why metanalysis are done, to bring conclusion over several species, experimental design and studies.

We agree with the reviewer about the limitations of meta-analysis, but our aim was to stress the still incomplete physiological characterizations of the coral response to OA, despite the large number of studies that have attempted to do it. The objective is to counterbalance the interpretation that physiological characterizations have limited capacity to investigate “the multiple interactions of environmental factors” that may explain the large variability documented for the OA response of symbiotic corals, as it is not possible to simulate natural variation in experimental tanks. We don't agree with this interpretation. On the contrary, we are convinced that a correct identification of the main targets and/or processes impacted by climate change stressors is fundamental for enhancing our capacity to predict the future of coral reefs, and, therefore, that physiological approaches are essential tools, particularly if they focus on the understanding of the mechanism affected.

line 289: Yes experimental analysis are required, and then can be combined within metanalysis to bring about general conclusion. These are not exclusive contrary to what it reads like here.

We agree with the reviewer that meta-analyses are complementary to experimental analysis in absence of the understanding of the biological mechanisms impacted. However, meta-analysis cannot recognize errors or inconsistencies in the literature data, and if the analysis is based on a large number of incomplete descriptions as well as erroneous interpretations, it is condemned to fail.

line 292: "the strength of the model system under analysis." i am not sure what is a "model system". The model system is the symbiotic coral. We changed this sentence by “*the level of knowledge of the system under analysis*” (line 296 in the revised manuscript).

line 298: see previous comment on light-stress As indicated previously, the variability of Fv/Fm is a powerful tool for monitoring increases in light pressure on photosystem II. This allows characterizing increases in the “excess” of energy absorbed by the photosynthetic membranes of the symbionts *in hospite*, in relation to their capacity to use this energy in carbon fixation. Therefore, reductions in Fv/Fm allow finding increases in the level of light-stress of the algae *in hospite* under similar external light environment.

line 305: see my previous comment on the grouping of the species, and therefore this conclusion on two different pathways for carbon allocation. I think it is not supported here.

The new PCA analysis allows a better support of our previous interpretation.

line 309: "Orbicella faveolata, also presents one of the largest adverse impacts on calcification" : but all measurements on *O. faveolata* show that there was no significant effect of OA...

As indicated previously, this response was found for the corals exposed to thermal-stress. Our aim in this sentence was to stress the inconsistency that the robust response to heat stress found for the photosynthetic metabolism of this species was however, not observed in calcification, as this species showed the largest reductions in calcification. It is interesting that the metabolic support of coral calcification for this species was not so affected by heat-stress, which suggests the need to understand other processes involved such as, for example, the metabolic link between photosynthesis/respiration and calcification.

Methods

line 332: The indications on how the carbonate chemistry was measured (and calculated) could be moved here.

Change performed as suggested. This description is now starting in line 337 of the new manuscript. We have also included a new table in the supplementary information (Table S5) with the variation of seawater chemistry in the OA treatment.

Experimental design: the author used only two tanks per treatment with 2 nubbins per species per tank. Many would consider that the true replication in this case is $n = 2$, but a recent guidelines stated that at least 2 tanks should be used for short term experiments (<7 days)... Moreover, in this study the temperature in the 2 tanks per treatment was regulated in a common header tank, same for OA. So the 2 tanks are not truly independent. In such case it could be concluded that the experiment is pseudoreplicated... Could the author provide an idea of what was the turnover of water in the header tank and the experimental tanks? As it is running water, it is possible that a high turnover would negate any potential tank effects.

The turnover rate of water in the tanks was quite high, about 1,5 minutes (90 s), with an average flow rate about 0.33 L s^{-1} . This information was documented by Vásquez-Elizondo and Enríquez (2016). Although we had indicated in the text that we followed a similar protocol, we have added this information in the revised manuscript to better clarify this important point to further readers.

line 383: Indicate how was the difference among days tested. If multiple t-tests were used, the alpha should be adjusted for multiple comparison.

The difference among days was tested for coral nubbins exposed to control conditions only employing individual t-tests for the studied physiological and optical parameters in order to confirm that they were exposed to the acclimation conditions. We did not employ alpha adjustment during the multiple testing of the effect of control condition on the physiological and optical parameters over time, based on the recommendation of Rubin 2021 as, in our case, we tested the individual null hypotheses of an effect on the parameters, and made decisions on each individual test. We did not undertake disjunction testing, in which case an adjustment of alpha levels is recommended according to this author (Rubin 2021) and Hulbert and Lombardi (2012). We are aware that the adjustment of alpha levels recommended by the reviewer, is more and more applied when employing multiple testing on the same data set to avoid type I errors. However, we also have taken into account that many authors also consider that those adjustments should be abandoned as they may lead to erroneous interpretations (O'Keefe 2003, Rothman 1990, see Hurlbert and Lombardi for a brief review, Rubin 2021).

Statistical analysis need to be revised. Except for a few case, the effect of $p\text{CO}_2$ is not significant in the 2 way anova, and only twice an interaction of pH and temperature is showed. Under this conditions, it is not advisable to conduct posthoc tests among individual cross treatments (except in the case of interaction). Moreover the authors used individual t tests for multiple comparison which is

really not appropriate (type I error). The author must revise the statistic and use more appropriate posthoc tests (Tukey HSD or Dunnett if the author only want to compare to control). Some data seems to have been transformed to validate the ANOVA assumptions but the transformations are not given, if it is too long at least these could be included in the supplementary table.

We have revised the statistics and provided a new table 2 with the posthoc HSD tests comparisons, and changed Figure 2 with the new significant differences found.

Figures

Figure 3: The panel a-h shows the same results but in a more complicated way than figure 2 in my opinion. Panel c and d are somewhat interesting but not really discussed in the manuscript. Panels i to l show results from a different study, and should be removed in my opinion as it only confuse the reader who is trying to understand the link with the current study.

We have removed panels i to l from figure 3, as suggested by the reviewer. The new figure 3 is limited to the former panels a-h, which we still found them useful to illustrate the moderate response of OA, panels a,b, relative to the severe and homogeneous response induced by thermal stress (e-h). We have also discussed in the manuscript panels c-d, which summarize these differences focusing on changes in P_{max} , G_{max} and symbiont density. In addition, a new figure 4 was added to illustrate the results of the PCA analysis, with the two groups identified and the averages \pm SE of the parameters that describe these differences. We thank the reviewer for his/her helpful suggestions, which has allowed us to better support the two type of response to OA identified. Future work, perhaps stimulate by our study, will elucidate the origin of these differences and the biological processes behind them.

Tables

Table 2: In addition to the fact that multiple t-tests cannot be used. Why is the degree of freedom different. It should be 6 but in some times, 5, 4 or even 2.17 ? This would indicate that some replicates were removed from the analysis and should be justified.

As indicated previously, table 2 has been replaced with a new table 2 that now includes the posthoc HSD tests comparisons among the treatments. The differences in degrees of freedom were due to two reasons: 1) some measured values, especially with the heat stressed samples, were below the detection limit, and thus excluded from the analysis; and 2) when the hypothesis Levene's test of homogeneity of variance was rejected, t and df were computed by SPSS applying the equal-variances-not-assumed degrees of freedom formula of the independent t-test as follows:

$$t = \frac{\bar{X}_1 - \bar{X}_2}{\sqrt{\frac{s_1^2}{n_1} + \frac{s_2^2}{n_2}}}$$

and,

$$df = \frac{\left(\frac{s_1^2}{n_1} + \frac{s_2^2}{n_2}\right)^2}{\frac{1}{n_1 - 1} \left(\frac{s_1^2}{n_1}\right)^2 + \frac{1}{n_2 - 1} \left(\frac{s_2^2}{n_2}\right)^2}$$

where

\bar{X}_1 = Mean of first sample

\bar{X}_2 = Mean of second sample

n_1 = Sample size of first sample

n_2 = Sample size of second sample

s_1 = Standard deviation of first sample

s_2 = Standard deviation of second sample

References:

- Anthony K. R., Kline D. I., Diaz-Pulido G., Dove S., Hoegh-Guldberg O. (2008). Ocean acidification causes bleaching and productivity loss in coral reef builders. *Proc. Natl. Acad. Sci. U.S.A.* 105, 17442-17446.
- Davies S. W., Marchetti A., Ries J. B., Castillo K. D. (2016) Thermal and pCO₂ stress elicit divergent transcriptomic responses in a resilient coral. *Front. Mar. Sci.* 3, 112.
- Gómez-Campo K., Enríquez S., Iglesias-Prieto R. (2022) A Road map for the development of the bleached coral phenotype. *Front. Mar. Sci.* 9, 806491.
- Hurlbert, S. H., & Lombardi, C. M. (2012). Lopsided reasoning on lopsided tests and multiple comparisons. *Australian & New Zealand Journal of Statistics*, 54, 23-42.
- O'Keefe, D. J. (2003). Colloquy: Should familywise alpha be adjusted? *Human Communication Research*, 29, 431-447.
- Rothman, K. J. (1990). No adjustments are needed for multiple comparisons. *Epidemiology*, 1, 43- 46.
- Rubin, M. (2021). When to adjust alpha during multiple testing: A consideration of disjunction, conjunction, and individual testing. *Synthese*.
- Scheufen T., Krämer W. E., Iglesias-Prieto R., Enríquez S. (2017a) Seasonal variation modulates coral sensibility to heat-stress and explains annual changes in coral productivity. *Sci. Rep.* 7, 4937.
- Scheufen T., Iglesias-Prieto R., Enríquez S. (2017b) Changes in the number of symbionts and Symbiodinium cell pigmentation modulate differentially coral light absorption and photosynthetic performance. *Front. Mar. Sci.* 4, 309.
- Suggett D. J., Dong, T L., Lawson F., Lawrenz E., Torres L., Smith D. J. (2013). Light availability determines susceptibility of reef building corals to ocean acidification. *Coral Reefs* 32, 327-337.
- Vásquez-Elizondo R. M., Enríquez S. (2016). Coralline algal physiology is more adversely affected by elevated temperature than reduced pH. *Sci. Rep.* 6, 19030.

REVIEWERS' COMMENTS:

Reviewer #2 (Remarks to the Author):

I have read the response to the reviewers' document, and I am satisfied that the authors have addressed each of my concerns and the concerns of the other reviewer. The manuscript has improved significantly. I still feel strongly about the short duration of the experiment, but the authors outlined some meaningful counterarguments. I support accepting this manuscript for publication.

Reviewer #3 (Remarks to the Author):

Thank you for the revisions. The manuscript was improved and I found the addition of the PCA in showing the different groups much more relevant than the previous grouping. I still have some comments before recommending this manuscript.

Abstract

Although the length is limited, I would suggest to add a sentence or even a few words highlighting the results and experiments that lead to the conclusion "on insufficient attention to key regulatory processes of these symbioses, particularly the metabolic dependence of coral calcification on algal photosynthesis and host respiration." At the moment, no reference to the experiment that was run is done.

Results

Fv/Fm (line 126 ~ 132) and line 387 of the method section: "Therefore, Fv/Fm was used to assess the accumulation of PSII photo-inactivation in the symbiotic algae, in hospite, due to photodamage." Following the author response, I am fine with the use of multiple t-test as the authors argue that the condition on each days are independent. But the use of t-test still need to be stated as in the revised manuscript it is still missing. Please add something like "Welch t-test were used to assess the difference in Fv/Fm between successive days"

Coral metabolic rates (line 133 ~ 188): t-tests are still reported, I though they have been replaced by tukey posthoc? Please clarify. Also I would suggest to use "Welch t-test" rather than "t-test" alone to avoid the previous misunderstanding where I though the authors used a "Student t-test".

line 151 for photosynthesis and 165 for calcification: "values not different from zero". I am sorry but I still do not understand the line of thought here that lead to the removal of zero values. In their previous response the author states that if the mean of the four replicate was not different from zero (how was that tested?) the positive or negative aspects of the value cannot be tested. I understand this but I do not see how it is relevant. For photosynthesis, the gross photosynthesis rates are showed, so negative values are not possible, therefore it is irrelevant. For calcification "net" calcification are reported. If net calcification is zero, it is either the "gross" calcification that is reduced or the dissolution that increases. Nevertheless, net calcification is still suppressed, no? Maybe the author wants to discuss the effect on "gross" calcification, ie the physiological process itself. But why removing some data when testing for the effect on Net calcification? and if some data were removed how many? this is not indicated in the method. I may misunderstand what the author are trying to show here.

Line 172: The addition of the PCA makes the results much easier to read.

discussion

Line 219 ~ 232: I found this paragraph "main focus" unclear. I think it is the importance of the resistance to light stress. However the first sentence is a little bit out of place or the importance of symbiont types for light stress resistance needs to be highlighted. In the sentence "our results, then, do not support [...]" on the relation between light and the impact of OA I would suggest to remove the "then" as I don't see the "immediate" relation with the previous sentence.

line 281: I find this paragraph quite strong against meta analysis. I agree with the author that good quality experimental studies are required, but it reads as because the study here is not in agreement with findings of metaanalysis, then meta analysis are "wrong". For example, I would suggest to tone down the sentence "As the capacity to predict the future of coral reefs depends on a correct identification of the main targets and/or processes impacted by climate change stressors, experimental analyses are the only means to isolate and thus elucidate both the individual impacts and possible interactions of environmental factors on organism performance". to something like "The capacity to predict the future of coral reefs depends on a correct identification of the main targets and/or processes impacted by climate change stressors, experimental analyses are essential in isolating and thus elucidating both the individual impacts and possible interactions of environmental factors on organism performance." Just removing the "only mean" makes the sentence softer. The previous sentence ending in "true threats to coral reefs" is also quite strong. For example Eyrie et al in their 2018 science paper, highlight the importance of sediment dissolution for the "net reef accretion" but do not conclude that it is the "true threat" and rather highlight in their conclusion that it is unknown whether reefs will experience catastrophic destruction or just slowly erode, with the later being a not so dramatic threats. But overall is this statement needed? The study here do not assess sediment dissolution but highlights that the effect of OA on coral calcification may not be as severe as previously thought. So why not simply rephrasing to something like: "Our study does not support these conclusions and show a minor effect of OA on coral calcification. Studies on carbonate dissolution, also show that rather than an impact on coral calcification, OA will affect coral reefs through an increase in net sediment dissolution67–69. "

Editor: Luke R. Grinham, PhD

As final revisions, please tone down some of the meta-analytical conclusions as suggested by Reviewer 3, and clearly state that your definition of "bleaching" differs from the classical definition.

Reviewer #2 (Remarks to the authors):

I have read the response to the reviewers' document, and I am satisfied that the authors have addressed each of my concerns and the concerns of the other reviewer. The manuscript has improved significantly. I still feel strongly about the short duration of the experiment, but the authors outlined some meaningful counterarguments. I support accepting this manuscript for publication.

We thank the reviewer for his/her support and positive comments.

Reviewer #3 (Remarks to the authors):

Thank you for the revisions. The manuscript was improved and I found the addition of the PCA in showing the different groups much more relevant than the previous grouping. I still have some comments before recommending this manuscript.

We thank the reviewer for his/her last comments.

Abstract

Although the length is limited, I would suggest to add a sentence or even a few words highlighting the results and experiments that lead to the conclusion "on insufficient attention to key regulatory processes of these symbioses, particularly the metabolic dependence of coral calcification on algal photosynthesis and host respiration." At the moment, no reference to the experiment that was run is done.

Due to format restrictions, we could not add the sentence indicated, but we consider that this conclusion maintained in the abstract may stimulate reading the whole manuscript in order to find the answers.

Results

Fv/Fm (line 126 ~ 132) and line 387 of the method section: "Therefore, Fv/Fm was used to assess the accumulation of PSII photo-inactivation in the symbiotic algae, in hospite, due to photodamage." Following the author response, I am fine with the use of multiple t-test as the authors argue that the condition on each days are independant. But the use of t-test still need to be stated as in the revised manuscript it is still missing. Please add something like "Welch t-test were used to assess the difference in Fv/Fm between successive days"

Thank you, we have added Welch t-test in supplementary table S3 to prevent this misunderstanding.

Coral metabolic rates (line 133 ~ 188): t-tests are still reported, I though they have been replaced by tukey posthoc? Please clarify. Tukey Posthoc test, as indicated in our previous answers, are in Table 2.

Also I would suggest to use "Welch t-test" rather than "t-test" alone to avoid the previous misunderstanding where I though the authors used a "Student t-test". Welch t-test are now indicated in the tests shown in table S3 from the supplementary information.

line 151 for photosynthesis and 165 for calcification: "values not different from zero". I am sorry but I still do not understand the line of thought here that lead to the removal of zero values. In their previous response the author states that if the mean of the four replicate was not different from zero (how was that tested?) the positive or negative aspects of the value cannot be tested. I understand this but I do not see how it is relevant. For photosynthesis, the gross photosynthesis rates are showed, so negative values are not possible, therefore it is irrelevant. For calcification "net" calcification are reported. If net calcification is zero, it is either the "gross" calcification that is reduced or the dissolution that increases. Nevertheless, net calcification is still suppressed, no? Maybe the author wants to discuss the effect on "gross" calcification, ie the physiological process itself. But why removing some data when testing for the effect on Net calcification? and if some data were removed how many? this is not indicated in the method. I may misunderstand what the author are trying to show here.

The physiological determinations (oxygen evolution or calcification) provide values which tend to zero, when it was not possible to distinguish the signal of the physiological activity from the basal noise of the electrical signal. Only under these conditions we considered that the physiological activity, oxygen evolution or calcification, did not occur. As we used four replicates for each determination, the calculation of the mean allowed finding positive values, only when this average showed a value above the error of our determinations. If not, the average value of the four replicates was considered Zero. No activity. In general, we did not find large discrepancies among samples. Even when the metabolic activity was low, in the stressed samples, we could determine positive low values for photosynthesis or calcification. However, when no oxygen evolution or no changes in the alkalinity anomaly were observed in most of the replicates, it was clear in these physiological determinations that the metabolic activity of the sample tended to zero. At least, the resolution of our physiological measurements was insufficient to detect this metabolic activity.

Line 172: The addition of the PCA makes the results much easier to read.

Thank you

discussion

Line 219 ~ 232: I found this paragraph "main focus" unclear. I think it is the importance of the resistance to light stress. However the first sentence is a little bit out of place or the importance of symbiont types for light stress resistance needs to be highlighted. In the sentence "our results, then, do not support [...]" on the relation between light and the impact of OA I would suggest to remove the "then" as I don't see the "immediate" relation with the previous sentence.

Thank you for pointing out this problem. The text has been corrected according to the reviewer suggestions. We changed the sentence indicating that we found "similar substantial impacts in all species..." despite the potential genetic variability of the dominant symbiont types. In addition the word, "then" has been removed from the sentence.

line 281: I find this paragraph quite strong against meta analysis. I agree with the author that good quality experimental studies are required, but it reads as because the study here is not in agreement with findings of metaanalysis, then meta analysis are "wrong". For example, I would suggest to tone down the sentence "As the capacity to predict the future of coral reefs depends on a correct identification of the main targets and/or processes impacted by climate change stressors, experimental analyses are the only means to isolate and thus elucidate both the individual impacts and possible interactions of environmental factors on organism performance". to something like "The capacity to predict the future of coral reefs depends on a correct identification of the main targets and/or processes impacted by climate change stressors, experimental analyses are essential in

isolating and thus elucidating both the individual impacts and possible interactions of environmental factors on

organism performance." Just removing the "only mean" makes the sentence softer. The previous sentence ending in "true threats to coral reefs" is also quite strong. For example Eyrie et al in their 2018 science paper, highlight the importance of sediment dissolution for the "net reef accretion" but do not conclude that it is the "true threat" and rather highlight in their conclusion that it is unknown whether reefs will experience catastrophic destruction or just slowly erode, with the later being a not so dramatic threats. But overall is this statement needed? The study here do not assess sediment dissolution but highlights that the effect of OA on coral calcification may not be as severe as previously thought. So why not simply rephrasing to something like: "Our study does not support these conclusions and show a minor effect of OA on coral calcification. Studies on carbonate dissolution, also show that rather than an impact on coral calcification, OA will affect coral reefs through an increase in net sediment dissolution^{67–69}. "

Text has been corrected according to the reviewer suggestions.